# Optimal Parallelization of Boosting

**Arthur da Cunha**
Department of Computer Science
Aarhus University
dac@cs.au.dk

**Mikael Møller Høgsgaard**
Department of Computer Science
Aarhus University
hogsgaard@cs.au.dk

**Kasper Green Larsen**
Department of Computer Science
Aarhus University
larsen@cs.au.dk

## Abstract

Recent works on the parallel complexity of Boosting have established strong lower bounds on the tradeoff between the number of training rounds $p$ and the total parallel work per round $t$. These works have also presented highly non-trivial parallel algorithms that shed light on different regions of this tradeoff. Despite these advancements, a significant gap persists between the theoretical lower bounds and the performance of these algorithms across much of the tradeoff space. In this work, we essentially close this gap by providing both improved lower bounds on the parallel complexity of weak-to-strong learners, and a parallel Boosting algorithm whose performance matches these bounds across the entire $p$ vs. $t$ compromise spectrum, up to logarithmic factors. Ultimately, this work settles the parallel complexity of Boosting algorithms that are nearly sample-optimal.

## 1 Introduction

Boosting is an extremely powerful and elegant idea that allows one to combine multiple inaccurate classifiers into a highly accurate *voting classifier*. Algorithms such as AdaBoost [Freund and Schapire, 1997] work by iteratively running a base learning algorithm on reweighted versions of the training data to produce a sequence of classifiers $h_1, \ldots, h_p$. After obtaining $h_i$, the weighting of the training data is updated to put larger weights on samples misclassified by $h_i$, and smaller weights on samples classified correctly. This effectively forces the next training iteration to focus on points with which the previous classifiers struggle. After sufficiently many rounds, the classifiers $h_1, \ldots, h_p$ are finally combined by taking a (weighted) majority vote among their predictions. Many Boosting algorithms have been developed over the years, for example Grove and Schuurmans [1998], Rätsch et al. [2005], Servedio [2003], Friedman [2001], with modern Gradient Boosting [Friedman, 2001] algorithms like XGBoost [Chen and Guestrin, 2016] and LightGBM [Ke et al., 2017] often achieving state-of-the-art performance on learning tasks while requiring little to no data cleaning. See e.g. the excellent survey by Natekin and Knoll [2013] for more background on Boosting.

While Boosting enjoys many advantages, it does have one severe drawback, also highlighted in Natekin and Knoll [2013]: Boosting is completely sequential as each of the consecutive training steps requires the output of previous steps to determine the reweighted learning problem. This property is shared by all Boosting algorithms and prohibits the use of computationally heavy training by the base learning algorithm in each iteration. For instance, Gradient Boosting algorithms often require hundreds to thousands of iterations to achieve the best accuracy. The crucial point is that even if you have access to thousands of machines for training, there is no way to parallelize the steps of Boosting and distribute the work among the machines (at least beyond the parallelization possible

38th Conference on Neural Information Processing Systems (NeurIPS 2024).

for the base learner). In effect, the training time of the base learning algorithm is directly multiplied by the number of steps of Boosting.

Multiple recent works [Long and Servedio, 2013, Karbasi and Larsen, 2024, Lyu et al., 2024] have studied parallelization of Boosting from a theoretical point of view, aiming for an understanding of the inherent tradeoffs between the number of training rounds $p$ and the total parallel work per round $t$. These works include both strong lower bounds on the cost of parallelization and highly non-trivial parallel Boosting algorithms with provable guarantees on accuracy. Previous studies however leave a significant gap between the performance of the parallel algorithms and the proven lower bounds.

The main contribution of this work is to close this gap by both developing a parallel algorithm with a better tradeoff between $p$ and $t$, as well as proving a stronger lower bound on this tradeoff. To formally state our improved results and compare them to previous works, we first introduce the theoretical framework under which parallel Boosting is studied.

**Weak-to-Strong Learning.**   Following the previous works Karbasi and Larsen [2024], Lyu et al. [2024], we study parallel Boosting in the theoretical setup of weak-to-strong learning. Weak-to-strong learning was introduced by Kearns [1988], Kearns and Valiant [1994] and has inspired the development of the first Boosting algorithms [Schapire, 1990]. In this framework, we consider binary classification over an input domain $\mathcal{X}$ with an unknown target concept $c\colon \mathcal{X} \to \{-1,1\}$ assigning labels to samples. A $\gamma$-weak learner for $c$ is then a learning algorithm $\mathcal{W}$ that for any distribution $\mathcal{D}$ over $\mathcal{X}$, when given at least some constant $m_0$ i.i.d. samples from $\mathcal{D}$, produces with constant probability a hypothesis $h$ with $\mathcal{L}_{\mathcal{D}}(h) \leq 1/2 - \gamma$. Here $\mathcal{L}_{\mathcal{D}}(h) = \Pr_{\mathbf{x}\sim\mathcal{D}}[h(\mathbf{x}) \neq c(\mathbf{x})]$. The goal in weak-to-strong learning is then to *boost* the accuracy of $\mathcal{W}$ by invoking it multiple times. Concretely, the aim is to produce a strong learner: A learning algorithm that for any distribution $\mathcal{D}$ over $\mathcal{X}$ and any $0 < \delta, \varepsilon < 1$, when given $m(\varepsilon, \delta)$ i.i.d. samples from $\mathcal{D}$, produces with probability at least $1 - \delta$ a hypothesis $h\colon \mathcal{X} \to \{-1,1\}$ such that $\mathcal{L}_{\mathcal{D}}(h) \leq \varepsilon$. We refer to $m(\varepsilon, \delta)$ as the *sample complexity* of the weak-to-strong learner.

Weak-to-strong learning has been extensively studied over the years, with many proposed algorithms, among which AdaBoost [Freund and Schapire, 1997] is perhaps the most famous. If $\mathcal{H}$ denotes a hypothesis set such that $\mathcal{W}$ always produces hypotheses from $\mathcal{H}$, and if $d$ denotes the VC-dimension of $\mathcal{H}$, then in terms of sample complexity, AdaBoost is known to produce a strong learner with sample complexity $m_{\mathrm{Ada}}(\varepsilon, \delta)$ satisfying

$$m_{\mathrm{Ada}}(\varepsilon, \delta) = O\left(\frac{d \ln(\frac{d}{\varepsilon\gamma}) \ln(\frac{1}{\varepsilon\gamma})}{\gamma^2 \varepsilon} + \frac{\ln(1/\delta)}{\varepsilon}\right). \tag{1}$$

This can be proved by observing that after $t = O(\gamma^{-2} \ln m)$ iterations, AdaBoost produces a voting classifier $f(x) = \mathrm{sign}(\sum_{i=1}^{t} \alpha_i h_i(x))$ with all *margins* on the training data being $\Omega(\gamma)$. The sample complexity bound then follows by invoking the best known generalization bounds for large margin voting classifiers [Breiman, 1999, Gao and Zhou, 2013]. Here the margin of the voting classifier $f$ on a training sample $(x, c(x))$ is defined as $c(x)\sum_{i=1}^{t} \alpha_i h_i(x) / \sum_{i=1}^{t} |\alpha_i|$. This sample complexity comes within logarithmic factors of the optimal sample complexity $m_{\mathrm{OPT}}(\varepsilon, \delta) = \Theta(d/(\gamma^2\varepsilon) + \ln(1/\delta)/\varepsilon)$ obtained e.g. in Larsen and Ritzert [2022].

**Parallel Weak-to-Strong Learning.**   The recent work by Karbasi and Larsen [2024] formalized parallel Boosting in the above weak-to-strong learning setup. Observing that all training happens in the weak learner, they proposed the following definition of parallel Boosting: A weak-to-strong learning algorithm has parallel complexity $(p, t)$ if for $p$ consecutive rounds it queries the weak learner with $t$ distributions. In each round $i$, if $D_1^i, \ldots, D_t^i$ denotes the distributions queried, the weak learner returns $t$ hypotheses $h_1^i, \ldots, h_t^i \in \mathcal{H}$ such that $\mathcal{L}_{D_j^i}(h_j^i) \leq 1/2 - \gamma$ for all $j$. At the end of the $p$ rounds, the weak-to-strong learner outputs a hypothesis $f\colon \mathcal{X} \to \{-1,1\}$. The queries made in each round and the final hypothesis $f$ must be computable from the training data as well as all hypotheses $h_j^i$ seen in previous rounds. The motivation for the above definition is that we could let one machine/thread handle each of the $t$ parallel query distributions in a round.

Since parallel weak-to-strong learning is trivial if we make no requirements on $\mathcal{L}_{\mathcal{D}}(f)$ for the output $f\colon \mathcal{X} \to \{-1,1\}$ (simply output $f(x) = 1$ for all $x \in \mathcal{X}$), we from hereon focus on parallel weak-to-strong learners that are near-optimal in terms of the sample complexity and accuracy tradeoff.

More formally, from the upper bound side, our goal is to obtain a sample complexity matching at least that of AdaBoost, stated in Eq. (1). That is, rewriting the loss $\varepsilon$ as a function of the number of samples $m$, we aim for output classifiers $f$ satisfying

$$\mathcal{L}_{\mathcal{D}}(f) = \mathrm{O}\left(\frac{d\ln(m)\ln(m/d) + \ln(1/\delta)}{\gamma^2 m}\right).$$

When stating lower bounds in the following, we have simplified the expressions by requiring that the *expected* loss satisfies $\mathcal{L}_{\mathcal{D}}(f) = \mathrm{O}(m^{-0.01})$. Note that this is far larger than the upper bounds, except for values of $m$ very close to $\gamma^{-2}d$. This only makes the lower bounds stronger. We remark that all the lower bounds are more general than this, but focusing on $m^{-0.01}$ in this introduction yields the cleanest bounds.

With these definitions, classic AdaBoost and other weak-to-strong learners producing voting classifiers with margins $\Omega(\gamma)$ all have a parallel complexity of $(\Theta(\gamma^{-2}\ln m), 1)$: They all need $\gamma^{-2}\ln m$ rounds to obtain $\Omega(\gamma)$ margins. Karbasi and Larsen [2024] presented the first alternative tradeoff by giving an algorithm with parallel complexity $(1, \exp(\mathrm{O}(d\ln(m)/\gamma^2)))$. Subsequent work by Lyu et al. [2024] gave a general tradeoff between $p$ and $t$. When requiring near-optimal accuracy, their tradeoff gives, for any $1 \leq R \leq 1/(2\gamma)$, a parallel complexity of $(\mathrm{O}(\gamma^{-2}\ln(m)/R), \exp(\mathrm{O}(dR^2))\ln(1/\gamma))$. The accuracy of both of these algorithms was proved by arguing that they produce a voting classifier with all margins $\Omega(\gamma)$.

On the lower bound side, Karbasi and Larsen [2024] showed that one of three things must hold: Either $p \geq \min\{\Omega(\gamma^{-1}\ln m), \exp(\Omega(d))\}$, or $t \geq \min\{\exp(\Omega(d\gamma^{-2})), \exp(\exp(\Omega(d)))\}$ or $p\ln(tp) = \Omega(d\ln(m)\gamma^{-2})$.

Lyu et al. [2024] also presented a lower bound that for some parameters is stronger than that of Karbasi and Larsen [2024], and for some is weaker. Concretely, they show that one of the following two must hold: Either $p \geq \min\{\Omega(\gamma^{-2}d), \Omega(\gamma^{-2}\ln m), \exp(\Omega(d))\}$, or $t \geq \exp(\Omega(d))$. Observe that the constraint on $t$ is only single-exponential in $d$, whereas the previous lower bound is double-exponential. On the other hand, the lower bound on $p$ is essentially stronger by a $\gamma^{-1}$ factor. Finally, they also give an alternative lower bound for $p = \mathrm{O}(\gamma^{-2})$, essentially yielding $p\ln t = \Omega(\gamma^{-2}d)$.

Even in light of the previous works, it is still unclear what the true complexity of parallel Boosting is. In fact, the upper and lower bounds only match in the single case where $p = \Omega(\gamma^{-2}\ln m)$ and $t = 1$, i.e. when standard AdaBoost is optimal.

**Our Contributions.** In this work, we essentially close the gap between the upper and lower bounds for parallel Boosting. From the upper bound side, we show the following general result.

**Theorem 1.1.** *Let $c\colon \mathcal{X} \to \{-1, 1\}$ be an unknown concept, $\mathcal{W}$ be a $\gamma$-weak learner for $c$ using a hypothesis set of VC-dimension $d$, $\mathcal{D}$ be an arbitrary distribution, and $\mathbf{S} \sim \mathcal{D}^m$ be a training set of size $m$. For all $R \in \mathbb{N}$, Algorithm 1 yields a weak-to-strong learner $\mathcal{A}_R$ with parallel complexity $(p, t)$ for*

$$p = \mathrm{O}\left(\frac{\ln m}{\gamma^2 R}\right) \qquad and \qquad t = e^{\mathrm{O}(dR)} \cdot \ln\frac{\ln m}{\delta\gamma^2},$$

*such that, with probability at least $1 - \delta$ over $\mathbf{S}$ and the randomness of $\mathcal{A}_R$, it holds that*

$$\mathcal{L}_{\mathcal{D}}(\mathcal{A}_R(\mathbf{S})) = \mathrm{O}\left(\frac{d\ln(m)\ln(m/d) + \ln(1/\delta)}{\gamma^2 m}\right).$$

Observe that this is a factor $R$ better than the bound by Lyu et al. [2024] in the exponent of $t$. Furthermore, if we ignore the $\ln(\ln(m)/(\delta\gamma^2))$ factor, it gives the clean tradeoff

$$p\ln t = \mathrm{O}\left(\frac{d\ln m}{\gamma^2}\right),$$

for any $p$ from 1 to $\mathrm{O}(\gamma^{-2}\ln m)$.

We complement our new upper bound by an essentially matching lower bound. Here we show that

**Theorem 1.2.** *There is a universal constant $C \geq 1$ for which the following holds. For any $0 < \gamma < 1/C$, any $d \geq C$, any sample size $m \geq C$, and any weak-to-strong learner $\mathcal{A}$ with parallel complexity $(p, t)$, there exists an input domain $\mathcal{X}$, a distribution $\mathcal{D}$, a concept $c \colon \mathcal{X} \to \{-1, 1\}$, and a $\gamma$-weak learner $\mathcal{W}$ for $c$ using a hypothesis set $\mathcal{H}$ of VC-dimension $d$ such that if the expected loss of $\mathcal{A}$ over the sample is no more than $m^{-0.01}$, then either $p \geq \min\{\exp(\Omega(d)), \Omega(\gamma^{-2} \ln m)\}$, or $t \geq \exp(\exp(\Omega(d)))$, or $p \ln t = \Omega(\gamma^{-2} d \ln m)$.*

Comparing Theorem 1.2 to known upper bounds, we first observe that $p = \Omega(\gamma^{-2} \ln m)$ corresponds to standard AdaBoost and is thus tight. The term $p = \exp(\Omega(d))$ is also near-tight. In particular, given $m$ samples, by Sauer-Shelah, there are only $O((m/d)^d) = \exp(O(d \ln(m/d)))$ distinct labellings by $\mathcal{H}$ on the training set. If we run AdaBoost, and in every iteration, we check whether a previously obtained hypothesis has advantage $\gamma$ under the current weighing, then we make no more than $\exp(O(d \ln(m/d)))$ queries to the weak learner (since every returned hypothesis must be distinct). The $p \ln t = \Omega(\gamma^{-2} d \ln m)$ matches our new upper bound in Theorem 1.1. Thus, only the $t \geq \exp(\exp(\Omega(d)))$ term does not match any known upper bound.

**Other Related Work.** Finally, we mention the work by Long and Servedio [2013], which initiated the study of the parallel complexity of Boosting. In their work, they proved that the parallel complexity $(p, t)$ must satisfy $p = \Omega(\gamma^{-2} \ln m)$, regardless of $t$ (they state it as $p = \Omega(\gamma^{-2})$, but it is not hard to improve by a $\ln m$ factor for loss $m^{-0.01}$). This seems to contradict the upper bounds above. The reason is that their lower bound has restrictions on which query distributions the weak-to-strong learner makes to the weak learner. The upper bounds above thus all circumvent these restrictions. As a second restriction, their lower bound instance has a VC-dimension that grows with $m$.

## 2 Upper Bound

In this section, we discuss our proposed method, Algorithm 1. Here, $C_{\mathrm{n}}$ refers a universal constant shared among results.

We provide a theoretical analysis of the algorithm, showing that it realizes the claims in Theorem 1.1. Our proof goes via the following intermediate theorem:

**Theorem 2.1.** *There exists universal constant $C_{\mathrm{n}} \geq 1$ such that for all $0 < \gamma < 1/2$, $R \in \mathbb{N}$, concept $c \colon \mathcal{X} \to \{-1, 1\}$, and hypothesis set $\mathcal{H} \subseteq \{-1, 1\}^{\mathcal{X}}$ of VC-dimension $d$, Algorithm 1 given an input training set $S \in \mathcal{X}^m$, a $\gamma$-weak learner $\mathcal{W}$,*

$$p \geq \frac{4 \ln m}{\gamma^2 R}, \qquad \text{and} \qquad t \geq e^{16 C_{\mathrm{n}} dR} \cdot R \ln \frac{pR}{\delta},$$

*produces a linear classifier $\mathbf{g}$ at Line 21 such that with probability at least $1 - \delta$ over the randomness of Algorithm 1, $\mathbf{g}(x) c(x) \geq \gamma/8$ for all $x \in S$.*

In Theorem 2.1 and throughout the paper, we define a *linear classifier* $g$ as linear combination of hypotheses $g(x) = \sum_{i=1}^{k} \alpha_i h_i(x)$ with $\sum_i |\alpha_i| = 1$. A linear classifier thus corresponds to a voting classifier with coefficients normalized and no sign operation. Observe that the voting classifier $f(x) = \mathrm{sign}(g(x))$ is correct if and only if $c(x) g(x) > 0$, where $c(x)$ is the correct label of $x$. Furthermore, $c(x) g(x)$ is the margin of the voting classifier $f$ on input $x$.

Theorem 1.1 follows from Theorem 2.1 via generalization bounds for linear classifiers with large margins. Namely, we apply Breiman's min-margin bound:

**Theorem 2.2** (Breiman [1999]). *Let $c \colon \mathcal{X} \to \{-1, 1\}$ be an unknown concept, $\mathcal{H} \subseteq \{-1, 1\}^{\mathcal{X}}$ a hypothesis set of VC-dimension $d$ and $\mathcal{D}$ an arbitrary distribution over $\mathcal{X}$. There is a universal constant $C > 0$ such that with probability at least $1 - \delta$ over a set of $m$ samples $\mathbf{S} \sim \mathcal{D}^m$, it holds for every linear classifier $g$ satisfying $c(x) g(x) \geq \gamma$ for all $(x, c(x)) \in \mathbf{S}$ that*

$$\mathcal{L}_{\mathcal{D}}(\mathrm{sign}(g)) \leq C \cdot \frac{d \ln(m) \ln(m/d) + \ln(1/\delta)}{\gamma^2 m}.$$

Thus far, our general strategy mirrors that of previous works: We seek to show that given suitable parameters Algorithm 1 produces a linear classifier with margins of order $\gamma$ with good probability.

---

**Algorithm 1:** Proposed parallel Boosting algorithm

> **Input** : Training set $S = \{(x_1, c(x_1)), \ldots, (x_m, c(x_m))\}$, $\gamma$-weak learner $\mathcal{W}$, number of calls to weak learner per round $t$, number of rounds $p$
> **Output:** Voting classifier $f$

**1** $\alpha \leftarrow \frac{1}{2} \ln \frac{1/2 + \gamma/2}{1/2 - \gamma/2}$
**2** $n \leftarrow C_{\mathrm{n}} d / \gamma^2$
**3** $D_1 \leftarrow (\frac{1}{m}, \frac{1}{m}, \ldots, \frac{1}{m})$
**4** **for** $k \leftarrow 0$ **to** $p - 1$ **do**
**5**    **parallel for** $r \leftarrow 1$ **to** $R$ **do**
**6**      **parallel for** $j \leftarrow 1$ **to** $t/R$ **do**
**7**        Sample $\mathbf{T}_{kR+r,j} \sim \boldsymbol{D}_{kR+1}^n$
**8**        $\mathbf{h}_{kR+r,j} \leftarrow \mathcal{W}(\mathbf{T}_{kR+r,j}, \mathrm{Uniform}(\mathbf{T}_{kR+r,j}))$
**9**      $\boldsymbol{\mathcal{H}}_{kR+r} \leftarrow \{\mathbf{h}_{kR+r,1}, \ldots, \mathbf{h}_{kR+r,t/R}\} \cup \{-\mathbf{h}_{kR+r,1}, \ldots, -\mathbf{h}_{kR+r,t/R}\}$
**10**    **for** $r \leftarrow 1$ **to** $R$ **do**
**11**      **if** *there exists* $\mathbf{h}^* \in \boldsymbol{\mathcal{H}}_{kR+r}$ *s.t.* $\mathcal{L}_{\boldsymbol{D}_{kR+r}}(\mathbf{h}^*) \leq 1/2 - \gamma/2$ **then**
**12**        $\mathbf{h}_{kR+r} \leftarrow \mathbf{h}^*$
**13**        $\alpha_{kR+r} \leftarrow \alpha$
**14**      **else**
**15**        $h_{kR+r} \leftarrow$ arbitrary hypothesis from $\boldsymbol{\mathcal{H}}_{kR+r}$
**16**        $\alpha_{kR+r} \leftarrow 0$
**17**      **for** $i \leftarrow 1$ **to** $m$ **do**
**18**        $\boldsymbol{D}_{kR+r+1}(i) \leftarrow \boldsymbol{D}_{kR+r}(i) \exp(-\boldsymbol{\alpha}_{kR+r} c(x_i) \mathbf{h}_{kR+r}(x_i))$
**19**      $\mathbf{Z}_{kR+r} \leftarrow \sum_{i=1}^m \boldsymbol{D}_{kR+r}(i) \exp(-\boldsymbol{\alpha}_{kR+r} c(x_i) \mathbf{h}_{kR+r}(x_i))$
**20**      $\boldsymbol{D}_{kR+r+1} \leftarrow \boldsymbol{D}_{kR+r+1} / \mathbf{Z}_{kR+r}$
**21** $\mathbf{g} \leftarrow x \mapsto \frac{1}{\sum_{j=1}^{pR} \boldsymbol{\alpha}_j} \sum_{j=1}^{pR} \boldsymbol{\alpha}_j \mathbf{h}_j(x)$
**22** **return** $\mathbf{f} \colon x \mapsto \mathrm{sign}(\mathbf{g}(x))$

---

Therefore, this section focuses on the lemmas that describe how, with suitable parameters, Algorithm 1 produces a classifier with large margins. With these results in hand, the proof of Theorem 2.1 becomes quite straightforward, so we defer it to Appendix B.3.

Algorithm 1 is a variant of Lyu et al. [2024, Algorithm 2]. The core idea is to use bagging to produce (in parallel) a set of hypotheses and use it to simulate a weak learner. To be more precise, we reason in terms of the following definition.

**Definition 1** ($\varepsilon$-approximation). Given a concept $c \colon \mathcal{X} \to \{-1, 1\}$, a hypothesis set $\mathcal{H} \subseteq \{-1, 1\}^{\mathcal{X}}$, and a distribution $\mathcal{D}$ over $\mathcal{X}$, a multiset $T$ is an *$\varepsilon$-approximation* for $\mathcal{D}$, $c$, and $\mathcal{H}$ if for all $h \in \mathcal{H}$, it holds that

$$|\mathcal{L}_{\mathcal{D}}(h) - \mathcal{L}_T(h)| \leq \varepsilon,$$

where $\mathcal{L}_T(h) \coloneqq \mathcal{L}_{\mathrm{Uniform}(T)}(h)$ is the empirical loss of $h$ on $T$. Moreover, we omit the reference to $c$ and $\mathcal{H}$ when no confusion seems possible.

Consider a reference distribution $D_0$ over a training dataset $S$. The bagging part of the method leverages the fact that if a subsample $\mathbf{T} \sim D_0^n$ is a $\gamma/2$-approximation for $D_0$, then inputting $\mathbf{T}$ (with the uniform distribution over it) to a $\gamma$-weak learner produces a hypothesis $h$ that, besides having advantage $\gamma$ on $\mathbf{T}$, also has advantage $\gamma/2$ on the entire dataset $S$ (relative to $D_0$). Indeed, in this setting, we have that $\mathcal{L}_{D_0}(h) \leq \mathcal{L}_{\mathbf{T}}(h) + \gamma/2 \leq 1/2 - \gamma + \gamma/2 = 1/2 - \gamma/2$. We can then take $h$ as if produced by a $\gamma/2$-weak learner queried with $(S, D_0)$, and compute a new distribution $D_1$ via a standard Boosting step[1]. That is, we can simulate a $\gamma/2$-weak learner as long as we can provide a $\gamma/2$-approximation for the target distribution. The strategy is to have a parallel bagging step in which we sample $\mathbf{T}_1, \mathbf{T}_2, \ldots, \mathbf{T}_t \overset{\mathrm{iid}}{\sim} D_0^n$ and query the $\gamma$-weak learner on each $\mathbf{T}_j$ to obtain hypotheses $\mathbf{h}_1, \ldots, \mathbf{h}_t$. Then, we search within these hypotheses to sequentially perform $R$ Boosting steps, obtaining distributions $D_1, D_2, \ldots, D_R$. As argued, this approach will succeed whenever we can

---

[1]Notice that we employ a fixed learning rate that assumes a worst-case advantage of $\gamma/2$.

find at least one $\gamma/2$-approximation for each $D_r$ among $\mathbf{h}_1, \mathbf{h}_2, \ldots, \mathbf{h}_t$. A single parallel round of querying the weak learner is thus sufficient for performing $R$ steps of Boosting, effectively reducing $p$ by a factor $R$. Crucially, testing the performance of the returned hypotheses $\mathbf{h}_1, \ldots, \mathbf{h}_t$ uses only inference/predictions and no calls to the weak learner.

The challenge is that the distributions $D_r$ diverge (exponentially fast) from $D_0$ as we progress in the Boosting steps. For the first Boosting step, the following classic result ensures a good probability of obtaining an approximation for $D_0$ when sampling from $D_0$ itself.

**Theorem 2.3** (Li et al. [2001], Talagrand [1994], Vapnik and Chervonenkis [1971])**.** *There is a universal constant $C > 0$ such that for any $0 < \varepsilon, \delta < 1$, $\mathcal{H} \subseteq \{-1, 1\}^{\mathcal{X}}$ of VC-dimension $d$, and distribution $\mathcal{D}$ over $\mathcal{X}$, it holds with probability at least $1 - \delta$ over a set $\mathbf{T} \sim \mathcal{D}^n$ that $\mathbf{T}$ is an $\varepsilon$-approximation for $\mathcal{D}$, $c$, and $\mathcal{H}$ provided that $n \geq C((d + \ln(1/\delta))/\varepsilon^2)$.*

However, we are interested in approximations for $D_r$ when we only have access to samples from $D_0$. Lyu et al. [2024] approaches this problem by tracking the "distance" between the distributions in terms of their *max-divergence*

$$\mathrm{D}_\infty(D_r, D_0) \coloneqq \ln\big(\sup_{x \in \mathcal{X}} D_r(x)/D_0(x)\big). \tag{2}$$

By bounding both $\mathrm{D}_\infty(D_r, D_0)$ and $\mathrm{D}_\infty(D_0, D_r)$, the authors can leverage the *advanced composition theorem* [Dwork et al., 2010][2] from the differential privacy literature to bound the probability of obtaining an approximation for $D_r$ when sampling from $D_0$. In turn, this allows them to relate the number of samples $t$ and the (sufficiently small) number of Boosting steps $R$ in a way that ensures a good probability of success at each step.

Besides setting up the application of advanced composition, the use of the max-divergence also simplifies the analysis since its "locality" allows one to bound the divergence between the two distributions via a worst-case study of a single entry. However, this approach sacrifices global information, limiting how much we can leverage our understanding of the distributions generated by Boosting algorithms. With that in mind, we instead track the distance between $D_r$ and $D_0$ in terms of the *Kullback-Leibler divergence* (KL divergence) [Kullback and Leibler, 1951] between them:

$$\mathrm{KL}(D_r \parallel D_0) \coloneqq \sum_{x \in \mathcal{X}} D_r(x) \ln \frac{D_r(x)}{D_0(x)}.$$

Comparing this expression to Eq. (2) reveals that the max-divergence is indeed a worst-case estimation of the KL divergence.

The KL divergence —also known as *relative entropy*— between two distributions $P$ and $Q$ is always non-negative and equal to zero if and only if $P = Q$. Moreover, in our setting, it is always finite due to the following remark.[3]

**Remark 1.** In the execution Algorithm 1, every distribution $D_\ell$, for $\ell \in [pR]$, has the same support. This must be the case since Line 20 always preserves the support of $D_1$.

On the other hand, the KL divergence is not a proper metric as it is not symmetric and it does not satisfy the triangle inequality, unlike the max-divergence. This introduces a number of difficulties in bounding the divergence between $\mathcal{D}_0$ and $\mathcal{D}_r$. Overcoming these challenges requires a deeper and highly novel analysis. Our results reveal that the KL divergence captures particularly well the behavior of our Boosting algorithm. We remark that we are not the first to relate KL divergence and Boosting, see e.g. Schapire and Freund [2012, Chapter 8 and the references therein], yet we make several new contributions to this connection.

To study the probability of obtaining a $\gamma/2$-approximation for $D_r$ when sampling from $D_0$, rather than using advanced composition, we employ the *duality formula for variational inference* [Donsker and Varadhan, 1975] —also known as *Gibbs variational principle*, or *Donsker-Varadhan formula*— to estimate such a probability in terms of $\mathrm{KL}(D_r \parallel D_0)$.

---

[2]Lemma 4.6 of Lyu et al. [2024].

[3]We only need $P$ to be absolutely continuous with respect to $Q$; i.e., that for any event $A$, we have $P(A) = 0$ whenever $Q(A) = 0$. We express our results in terms of identical supports for the sake of simplicity as they can be readily generalized to only require absolute continuity.

**Lemma 2.4** (Duality formula[4])**.** *Given finite probability spaces $(\Omega, \mathcal{F}, P)$ and $(\Omega, \mathcal{F}, Q)$, if $P$ and $Q$ have the same support, then for any real-valued random variable $\mathbf{X}$ on $(\Omega, \mathcal{F}, P)$ we have that*

$$\ln \mathbb{E}_P\left[e^{\mathbf{X}}\right] \geq \mathbb{E}_Q[\mathbf{X}] - \mathrm{KL}(Q \parallel P). \tag{3}$$

Lemma 2.4 allows us to prove that if $\mathrm{KL}(D_r \parallel D_0)$ is sufficiently small, then the probability of obtaining a $\gamma/2$-approximation for $D_r$ when sampling from $D_0$ is sufficiently large. Namely, we prove the following.

**Lemma 2.5.** *There exists universal constant $C_{\mathrm{n}} \geq 1$ for which the following holds. Given $0 < \gamma < 1/2$, $R, m \in \mathbb{N}$, concept $c\colon \mathcal{X} \to \{-1, 1\}$, and hypothesis set $\mathcal{H} \subseteq \{-1, 1\}^{\mathcal{X}}$ of VC-dimension $d$, let $\tilde{D}$ and $D$ be distributions over $[m]$ and $\mathcal{G} \in [m]^*$ be the family of $\gamma/2$-approximations for $D$, $c$, and $\mathcal{H}$. If $\tilde{D}$ and $D$ have the same support and*

$$\mathrm{KL}(D \parallel \tilde{D}) \leq 4\gamma^2 R,$$

*then for all $n \geq C_{\mathrm{n}} \cdot d/\gamma^2$ it holds that*

$$\Pr_{\mathbf{T} \sim \tilde{D}^n}[\mathbf{T} \in \mathcal{G}] \geq \exp(-16 C_{\mathrm{n}} dR).$$

*Proof sketch.* Our argument resembles a proof of the Chernoff bound: After taking exponentials on both sides of Eq. (3), we exploit the generality of Lemma 2.4 by defining the random variable $\mathbf{X}\colon T \mapsto \lambda \mathbf{1}_{\{T \in \mathcal{G}\}}$ and later carefully choosing $\lambda$. We then note that Theorem 2.3 ensures that $\mathbf{X}$ has high expectation for $\mathbf{T} \sim D^n$. Setting $\lambda$ to leverage this fact, we obtain a lower bound on the expectation of $\mathbf{X}$ relative to $\mathbf{T} \sim \tilde{D}^n$, yielding the thesis. $\qquad\square$

We defer the detailed proof to Appendix B.1.

With Lemma 2.5 in hand, recall that our general goal is to show that, with high probability, the linear classifier $g$ produced by Algorithm 1 satisfies that $c(x)g(x) = \Omega(\gamma)$ for all $x \in S$. Standard techniques allow us to further reduce this goal to that of showing that the product of the normalization factors, $\prod_{\ell=1}^{pR} Z_\ell$, is sufficiently small. Accordingly, in our next lemma, we bound the number of samples needed in the bagging step to obtain a small product of the normalization factors produced by the Boosting steps.

Here, the analysis in terms of the KL divergence delivers a clear insight into the problem, revealing an interesting trichotomy: if $\mathrm{KL}(D_r \parallel D_0)$ is small, Lemma 2.5 yields the result; on the other hand, if $D_r$ has diverged too far from $D_0$, then either the algorithm has already made enough progress for us to skip a step, or the negation of some hypothesis used in a previous step has sufficient advantage relative to the distribution at hand. Formally, we prove the following.

**Lemma 2.6.** *There exists universal constant $C_{\mathrm{n}} \geq 1$ such that for all $R \in \mathbb{N}$, $0 < \delta < 1$, $0 < \gamma < 1/2$, and $\gamma$-weak learner $\mathcal{W}$ using a hypothesis set $\mathcal{H} \subseteq \{-1, 1\}^{\mathcal{X}}$ with VC-dimension $d$, if $t \geq R \cdot \exp(16 C_{\mathrm{n}} dR) \cdot \ln(R/\delta)$, then with probability at least $1 - \delta$ the hypotheses $\mathbf{h}_{kR+1}, \ldots, \mathbf{h}_{kR+R}$ obtained by Algorithm 1 induce normalization factors $\mathbf{Z}_{kR+1}, \ldots, \mathbf{Z}_{kR+R}$ such that*

$$\prod_{r=1}^{R} \mathbf{Z}_{kR+r} < \exp(-\gamma^2 R/2).$$

*Proof sketch.* We assume for simplicity that $k = 0$ and argue by induction on $R' \in [R]$. After handling the somewhat intricate stochastic relationships of the problem, we leverage the simple

---

[4]Corollary of, e.g., Dembo and Zeitouni [1998, Lemma 6.2.13] or Lee [2022, Theorem 2.1]. Presented here in a weaker form for the sake of simplicity.

remark that $\mathrm{KL}(\boldsymbol{D}_{R'} \| \boldsymbol{D}_{R'}) = 0$ to reveal the following telescopic decomposition:

$$
\begin{aligned}
\mathrm{KL}(\boldsymbol{D}_{R'} \| \boldsymbol{D}_1) &= \mathrm{KL}(\boldsymbol{D}_{R'} \| \boldsymbol{D}_1) - \mathrm{KL}(\boldsymbol{D}_{R'} \| \boldsymbol{D}_{R'}) \\
&= \mathrm{KL}(\boldsymbol{D}_{R'} \| \boldsymbol{D}_1) - \mathrm{KL}(\boldsymbol{D}_{R'} \| \boldsymbol{D}_2) \\
&\quad + \mathrm{KL}(\boldsymbol{D}_{R'} \| \boldsymbol{D}_2) - \mathrm{KL}(\boldsymbol{D}_{R'} \| \boldsymbol{D}_3) \\
&\quad + \cdots \\
&\quad + \mathrm{KL}(\boldsymbol{D}_{R'} \| \boldsymbol{D}_{R'-1}) - \mathrm{KL}(\boldsymbol{D}_{R'} \| \boldsymbol{D}_{R'}) \\
&= \sum_{r=1}^{R'-1} \mathrm{KL}(\boldsymbol{D}_{R'} \| \boldsymbol{D}_r) - \mathrm{KL}(\boldsymbol{D}_{R'} \| \boldsymbol{D}_{r+1}).
\end{aligned}
$$

Moreover, given $r \in \{1, \ldots, R'-1\}$,

$$
\begin{aligned}
\mathrm{KL}(\boldsymbol{D}_{R'} \| \boldsymbol{D}_r) - \mathrm{KL}(\boldsymbol{D}_{R'} \| \boldsymbol{D}_{r+1}) &= \sum_{i=1}^{m} \boldsymbol{D}_{R'}(i) \ln \frac{\boldsymbol{D}_{R'}(i)}{\boldsymbol{D}_r(i)} - \sum_{i=1}^{m} \boldsymbol{D}_{R'}(i) \ln \frac{\boldsymbol{D}_{R'}(i)}{\boldsymbol{D}_{r+1}(i)} \\
&= \sum_{i=1}^{m} \boldsymbol{D}_{R'}(i) \ln \frac{\boldsymbol{D}_{r+1}(i)}{\boldsymbol{D}_r(i)} \\
&= -\ln \mathbf{Z}_r - \sum_{i=1}^{m} \boldsymbol{D}_{R'}(i) \boldsymbol{\alpha}_r c(x_i) \mathbf{h}_r(x_i).
\end{aligned}
$$

Altogether, we obtain that

$$
\mathrm{KL}(\boldsymbol{D}_{R'} \| \boldsymbol{D}_1) = -\ln \prod_{r=1}^{R'-1} \mathbf{Z}_r + \sum_{r=1}^{R'-1} \boldsymbol{\alpha}_r \sum_{i=1}^{m} \boldsymbol{D}_{R'}(i) c(x_i)(-\mathbf{h}_r(x_i)).
$$

Now, if $\mathrm{KL}(\boldsymbol{D}_{R'} \| \boldsymbol{D}_1)$ is small (at most $4\gamma^2 R$), Lemma 2.5 ensures that with sufficient probability there exists a $\gamma/2$-approximation for $\boldsymbol{D}_{R'}$ within $\mathbf{T}_{R',1}, \ldots, \mathbf{T}_{R',t/R}$, yielding the induction step (by Claim 1). Otherwise, if $\mathrm{KL}(\boldsymbol{D}_{R'} \| \boldsymbol{D}_1)$ is large, then either *(i)* the term $-\ln \prod_{r=1}^{R'-1} \mathbf{Z}_r$ is large enough for us to conclude that $\prod_{r=1}^{R'-1} \mathbf{Z}_r$ is already less than $\exp(-\gamma^2 R'/2)$ and we can skip the step; or *(ii)* the term $\sum_{r=1}^{R'-1} \boldsymbol{\alpha}_r \sum_{i=1}^{m} \boldsymbol{D}_{R'}(i) c(x_i)(-\mathbf{h}_r(x_i))$ is sufficiently large to imply the existence of $\mathbf{h}^* \in \{-\mathbf{h}_1, \ldots, -\mathbf{h}_{R'-1}\}$ satisfying that

$$
\sum_{i=1}^{m} \boldsymbol{D}_{R'}(i) c(x_i) \mathbf{h}^*(x_i) > \gamma,
$$

which implies that such $\mathbf{h}^*$ has margin at least $\gamma$ with respect to $\boldsymbol{D}_{R'}$ and we can conclude the induction step as before. $\qquad \square$

We defer the detailed proof to Appendix B.2.

## 3 Overview of the Lower Bound

In this section, we provide an overview of the main ideas behind our improved lower bound. The details are available in Appendix C.

Our lower bound proof is inspired by and builds upon the work of Lyu et al. [2024], which we now summarize. Similarly to Karbasi and Larsen [2024], they consider an input domain $\mathcal{X} = [2m]$, where $m$ denotes the number of training samples available for a weak-to-strong learner $\mathcal{A}$ with parallel complexity $(p, t)$. In their construction, they consider a uniform random concept $\mathbf{c} \colon \mathcal{X} \to \{-1, 1\}$ and give a randomized construction of a weak learner. Proving a lower bound on the expected error of $\mathcal{A}$ under this random choice of concept and weak learner implies, by averaging, the existence of a deterministic choice of concept and weak learner for which $\mathcal{A}$ has at least the same error.

The weak learner is constructed by drawing a random hypothesis set $\mathcal{H}$, using inspiration from the so-called *coin problem*. In the coin problem, we observe $p$ independent outcomes of a biased coin

and the goal is to determine the direction of the bias. If a coin has a bias of $\beta$, then upon seeing $n$ outcomes of the coin, any algorithm for guessing the bias of the coin is wrong with probability at least $\exp(-\mathrm{O}(\beta^2 n))$. Now to connect this to parallel Boosting, Lyu et al. construct $\mathcal{H}$ by adding $\mathbf{c}$ as well as $p$ random hypotheses $\mathbf{h}_1, \ldots, \mathbf{h}_p$ to $\mathcal{H}$. Each hypothesis $\mathbf{h}_i$ has each $\mathbf{h}_i(x)$ chosen independently with $\mathbf{h}_i(x) = \mathbf{c}(x)$ holding with probability $1/2 + 2\gamma$. The weak learner $\mathcal{W}$ now processes a query distribution $D$ by returning the first hypothesis $\mathbf{h}_i$ with advantage $\gamma$ under $D$. If no such hypothesis exists, it instead returns $\mathbf{c}$. The key observation is that if $\mathcal{W}$ is never forced to return $\mathbf{c}$, then the only information $\mathcal{A}$ has about $\mathbf{c}(x)$ for each $x$ not in the training data (which is at least half of all $x$, since $|\mathcal{X}| = 2m$), is the outcomes of up to $p$ coin tosses that are $2\gamma$ biased towards $\mathbf{c}(x)$. Thus, the expected error becomes $\exp(-\mathrm{O}(\gamma^2 p))$. For this to be smaller than $m^{-0.01}$ then requires $p = \Omega(\gamma^{-2}\ln m)$ as claimed in their lower bound.

The last step of their proof, is then to argue that $\mathcal{W}$ rarely has to return $\mathbf{c}$ upon a query. The idea here is to show that in the $i$th parallel round, $\mathcal{W}$ can use $\mathbf{h}_i$ to answer all queries, provided that $t$ is small enough. This is done by observing that for any query distribution $D$ that is independent of $\mathbf{h}_i$, the *expected* loss satisfies $\mathbb{E}_{\mathbf{h}_i}[\mathcal{L}_D(\mathbf{h}_i)] = 1/2 - 2\gamma$ due to the bias. Using inspiration from Karbasi and Larsen [2024], they then show that for sufficiently "well-spread" queries $D$, the loss of $\mathbf{h}_i$ under $D$ is highly concentrated around its expectation (over the random choice of $\mathbf{h}_i$) and thus $\mathbf{h}_i$ may simultaneously answer all (up to) $t$ well-spread queries in round $i$. To handle "concentrated" queries, i.e., query distribution with most of the weight on a few $x$, they also use ideas from Karbasi and Larsen [2024] to argue that if we add $2^{\mathrm{O}(d)}$ uniform random hypotheses to $\mathcal{H}$, then these may be used to answer all concentrated queries.

Note that the proof crucially relies on $\mathbf{h}_i$ being independent of the queries in the $i$th round. Here the key idea is that if $\mathcal{W}$ can answer all the queries in round $i$ using $\mathbf{h}_i$, then $\mathbf{h}_{i+1}, \ldots, \mathbf{h}_p$ are independent of any queries the weak-to-strong learner makes in round $i + 1$.

In our improved lower bound, we observe that the expected error of $\exp(-\mathrm{O}(\gamma^2 p))$ is much larger than $m^{-0.01}$ for small $p$. That is, the previous proof is in some sense showing something much too strong when trying to understand the tradeoff between $p$ and $t$. What this gives us, is that we can afford to make the coins/hypotheses $\mathbf{h}_i$ much more biased towards $\mathbf{c}$ when $p$ is small. Concretely, we can let the bias be as large as $\beta = \Theta(\sqrt{\ln(m)/p})$, which may be much larger than $2\gamma$. This, in turn, makes it significantly more likely that $\mathbf{h}_i$ can answer an independently chosen query distribution $D$. In this way, the same $\mathbf{h}_i$ may answer a much larger number of queries $t$, resulting in a tight tradeoff between the parameters. As a second contribution, we also find a better way of analyzing this lower bound instance, improving one term in the lower bound on $t$ from $\exp(\Omega(d))$ to $\exp(\exp(d))$. We refer the reader to the full proof for details.

# 4 Conclusion

In this paper, we have addressed the parallelization of Boosting algorithms. By establishing both improved lower bounds and an essentially optimal algorithm, we have effectively closed the gap between theoretical lower bounds and performance guarantees across the entire tradeoff spectrum between the number of training rounds and the parallel work per round.

Given that, we believe future work may focus on better understanding the applicability of the theoretical tools developed here to other settings since some lemmas obtained seem quite general. They may aid, for example, in investigating to which extent the post-processing of hypotheses obtained in the bagging step can improve the complexity of parallel Boosting algorithms, which remains as an interesting research direction.

## Acknowledgments and Disclosure of Funding

This research is co-funded by the European Union (ERC, TUCLA, 101125203) and Independent Research Fund Denmark (DFF) Sapere Aude Research Leader Grant No. 9064-00068B. Views and opinions expressed are however those of the author(s) only and do not necessarily reflect those of the European Union or the European Research Council. Neither the European Union nor the granting authority can be held responsible for them.

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

# A    Auxiliary Results

In this section we state and proof claims utilized in our argument. The arguments behind those are fairly standard, so they are not explicitly stated in the main text.

**Claim 1.** *Let $\ell \in \mathbb{N}$ and $0 < \gamma < 1/2$. If a hypothesis $h_\ell$ has advantage $\gamma_\ell$ satisfying $\mathcal{L}_{D_\ell}(h_\ell) = 1/2 - \gamma_\ell \leq 1/2 - \gamma/2$ and $\alpha_\ell = \alpha$, then*

$$Z_\ell \leq \sqrt{1 - \gamma^2} \leq e^{-\gamma^2/2}.$$

*Proof.* It holds that

$$
\begin{aligned}
Z_\ell &= \sum_{i=1}^{m} D_\ell(i) \exp(-\alpha_\ell c(x_i) h_\ell(x_i)) \\
&= \sum_{i:h_\ell(x_i)=c(x_i)} D_\ell(i) e^{-\alpha} + \sum_{i:h_\ell(x_i)\neq c(x_i)} D_\ell(i) e^{\alpha} \\
&= \left(\frac{1}{2} + \gamma_\ell\right) \sqrt{\frac{1-\gamma}{1+\gamma}} + \left(\frac{1}{2} - \gamma_\ell\right) \cdot \sqrt{\frac{1+\gamma}{1-\gamma}} \\
&= \left(\frac{1/2 + \gamma_\ell}{1+\gamma} + \frac{1/2 - \gamma_\ell}{1-\gamma}\right) \sqrt{(1+\gamma)(1-\gamma)} \\
&= \left(\frac{1 - 2\gamma \cdot \gamma_\ell}{1 - \gamma^2}\right) \sqrt{1 - \gamma^2}.
\end{aligned}
$$

Finally, since $\gamma_\ell \geq \gamma/2$ and $\gamma \in (0, 1/2)$, and, thus, $1 - \gamma^2 > 0$, we have that

$$\frac{1 - 2\gamma \cdot \gamma_\ell}{1 - \gamma^2} \leq \frac{1 - \gamma^2}{1 - \gamma^2} = 1.$$

□

**Claim 2.** *Algorithm 1 produces a linear classifier $\mathbf{g}$ whose exponential loss satisfies*

$$\sum_{i=1}^{m} \exp\left(-c(x_i) \sum_{j=1}^{pR} \boldsymbol{\alpha}_j \mathbf{h}_j(x_i)\right) = m \prod_{j=1}^{pR} \mathbf{Z}_j.$$

*Proof.* It suffices to consider the last distribution $\boldsymbol{D}_{pR+1}$ produced by the algorithm. It holds that

$$
\begin{aligned}
1 &= \sum_{i=1}^{m} \boldsymbol{D}_{pR+1}(i) && \text{(as } \boldsymbol{D}_{pR+1} \text{ is a distribution)} \\
&= \sum_{i=1}^{m} \boldsymbol{D}_{pR}(i) \cdot \frac{\exp(-\boldsymbol{\alpha}_{pR} c(x_i) \mathbf{h}_{pR}(x_i))}{\mathbf{Z}_{pR}} && \text{(by Line 20)} \\
&= \sum_{i=1}^{m} D_1(i) \cdot \prod_{j=1}^{pR} \frac{\exp(-\boldsymbol{\alpha}_j c(x_i) \mathbf{h}_j(x_i))}{\mathbf{Z}_j} && \text{(by further unrolling the } \boldsymbol{D}_j\text{s)} \\
&= \frac{1}{m} \cdot \sum_{i=1}^{m} \frac{\exp(-c(x_i) \sum_{j=1}^{pR} \boldsymbol{\alpha}_j \mathbf{h}_j(x_i))}{\prod_{j=1}^{pR} \mathbf{Z}_j}. && \text{(as } D_1 \text{ is uniform)}
\end{aligned}
$$

□

# B    Detailed Proofs

In this section, provide full proofs for the results from Section 2. For convenience, we provide copies of the statements before each proof.

## B.1 Proof of Lemma 2.5

**Lemma 2.5.** *There exists universal constant $C_\mathrm{n} \geq 1$ for which the following holds. Given $0 < \gamma < 1/2$, $R, m \in \mathbb{N}$, concept $c\colon \mathcal{X} \to \{-1, 1\}$, and hypothesis set $\mathcal{H} \subseteq \{-1, 1\}^{\mathcal{X}}$ of VC-dimension $d$, let $\tilde{D}$ and $D$ be distributions over $[m]$ and $\mathcal{G} \in [m]^*$ be the family of $\gamma/2$-approximations for $D$, $c$, and $\mathcal{H}$. If $\tilde{D}$ and $D$ have the same support and*

$$\mathrm{KL}(D \parallel \tilde{D}) \leq 4\gamma^2 R,$$

*then for all $n \geq C_\mathrm{n} \cdot d/\gamma^2$ it holds that*

$$\Pr_{\mathbf{T} \sim \tilde{D}^n}[\mathbf{T} \in \mathcal{G}] \geq \exp(-16 C_\mathrm{n} dR).$$

*Proof.* Let $\lambda \in \mathbb{R}_{>0}$ (to be chosen later) and $\mathbf{X}\colon [m]^n \to \{0, \lambda\}$ be the random variable given by

$$\mathbf{X}(T) = \lambda \mathbf{1}_{\{T \in \mathcal{G}\}}.$$

Since $\tilde{D}$ and $D$ have the same support, so do $\tilde{D}^n$ and $D^n$. Thus, taking the exponential of both sides of Eq. (3), Lemma 2.4 yields that

$$\exp(-\mathrm{KL}(D^n \parallel \tilde{D}^n) + \mathbb{E}_{D^n}[\mathbf{X}]) \leq \mathbb{E}_{\tilde{D}^n}[e^{\mathbf{X}}]. \tag{4}$$

We have that

$$\mathbb{E}_{D^n}[\mathbf{X}] = \lambda \cdot \Pr_{\mathbf{T} \sim D^n}[\mathbf{T} \in \mathcal{G}]. \tag{5}$$

Moreover,

$$\begin{aligned}
\mathbb{E}_{\tilde{D}^n}[e^{\mathbf{X}}] &= \mathbb{E}_{\mathbf{T} \sim \tilde{D}^n}[e^\lambda \cdot \mathbf{1}_{\{\mathbf{T} \in \mathcal{G}\}} + \mathbf{1}_{\{\mathbf{T} \notin \mathcal{G}\}}] \\
&= \mathbb{E}_{\mathbf{T} \sim \tilde{D}^n}[e^\lambda \cdot \mathbf{1}_{\{\mathbf{T} \in \mathcal{G}\}} + 1 - \mathbf{1}_{\{\mathbf{T} \in \mathcal{G}\}}] \\
&= 1 + (e^\lambda - 1)\mathbb{E}_{\mathbf{T} \sim \tilde{D}^n}[\mathbf{1}_{\{\mathbf{T} \in \mathcal{G}\}}] \\
&= 1 + (e^\lambda - 1)\Pr_{\mathbf{T} \sim \tilde{D}^n}[\mathbf{T} \in \mathcal{G}]. \tag{6}
\end{aligned}$$

Applying Eqs. (5) and (6) to Eq. (4), we obtain that

$$\exp\left(-\mathrm{KL}(D^n \parallel \tilde{D}^n) + \lambda \Pr_{\mathbf{T} \sim D^n}[\mathbf{T} \in \mathcal{G}]\right) \leq 1 + (e^\lambda - 1)\Pr_{\mathbf{T} \sim \tilde{D}^n}[\mathbf{T} \in \mathcal{G}]$$

and, thus,

$$\Pr_{\mathbf{T} \sim \tilde{D}^n}[\mathbf{T} \in \mathcal{G}] \geq \frac{\exp\left[-\mathrm{KL}(D^n \parallel \tilde{D}^n) + \lambda \Pr_{\mathbf{T} \sim D^n}[\mathbf{T} \in \mathcal{G}]\right] - 1}{e^\lambda - 1}$$

for any $\lambda > 0$. Choosing

$$\lambda = \frac{\mathrm{KL}(D^n \parallel \tilde{D}^n) + \ln 2}{\Pr_{\mathbf{T} \sim D^n}[\mathbf{T} \in \mathcal{G}]},$$

we obtain that

$$\begin{aligned}
\Pr_{\mathbf{T} \sim \tilde{D}^n}[\mathbf{T} \in \mathcal{G}] &\geq \frac{1}{e^\lambda - 1} \\
&\geq e^{-\lambda}. \tag{7}
\end{aligned}$$

Now, by Theorem 2.3 (using $\delta = 1/2$), there exists a constant $C_\mathrm{n} \geq 1$ such that having

$$n \geq C_\mathrm{n} \cdot \frac{d}{\gamma^2}$$

ensures that

$$\Pr_{\mathbf{T} \sim D^n}[\mathbf{T} \in \mathcal{G}] \geq \frac{1}{2}.$$

Also, since, by hypothesis, $\mathrm{KL}(D \parallel \tilde{D}) \leq 4\gamma^2 R$, we have that

$$\mathrm{KL}(D^n \parallel \tilde{D}^n) = n \, \mathrm{KL}(D \parallel \tilde{D})$$
$$\leq 4C_{\mathrm{n}} dR.$$

Applying it to Eq. (7), we conclude that

$$\Pr_{\mathbf{T} \sim \tilde{D}^n}[\mathbf{T} \in \mathcal{G}] \geq \exp\left(-\frac{4C_{\mathrm{n}} dR + \ln 2}{1/2}\right)$$
$$\geq \exp(-16C_{\mathrm{n}} dR).$$

$\square$

## B.2 Proof of Lemma 2.6

**Lemma 2.6.** *There exists universal constant $C_{\mathrm{n}} \geq 1$ such that for all $R \in \mathbb{N}$, $0 < \delta < 1$, $0 < \gamma < 1/2$, and $\gamma$-weak learner $\mathcal{W}$ using a hypothesis set $\mathcal{H} \subseteq \{-1, 1\}^{\mathcal{X}}$ with VC-dimension $d$, if $t \geq R \cdot \exp(16C_{\mathrm{n}} dR) \cdot \ln(R/\delta)$, then with probability at least $1 - \delta$ the hypotheses $\mathbf{h}_{kR+1}, \dots, \mathbf{h}_{kR+R}$ obtained by Algorithm 1 induce normalization factors $\mathbf{Z}_{kR+1}, \dots, \mathbf{Z}_{kR+R}$ such that*

$$\prod_{r=1}^{R} \mathbf{Z}_{kR+r} < \exp(-\gamma^2 R/2).$$

*Proof.* Assume, for simplicity, that $k = 0$.

Letting

$$\mathcal{E}_{R'} = \left\{\prod_{r=1}^{R'} \mathbf{Z}_r < \exp(-\gamma^2 R'/2)\right\},$$

we will show that for all $R' \in [R]$ it holds that

$$\Pr[\mathcal{E}_{R'} \mid \mathcal{E}_1, \dots, \mathcal{E}_{R'-1}] \geq 1 - \delta/R. \tag{8}$$

The thesis then follows by noting that

$$\Pr[\mathcal{E}_1 \cap \cdots \cap \mathcal{E}_R] = \prod_{r=1}^{R} \Pr[\mathcal{E}_r \mid \mathcal{E}_1, \dots, \mathcal{E}_{r-1}] \qquad \text{(by the chain rule)}$$
$$\geq \left(1 - \frac{\delta}{R}\right)^R \qquad \text{(by Eq. (8))}$$
$$\geq 1 - R \cdot \frac{\delta}{R} \qquad \text{(by Bernoulli's inequality)}$$
$$= 1 - \delta.$$

Let $\mathcal{G}_{\boldsymbol{D}_{R'}} \subseteq [m]^n$ be the family of $\gamma/2$-approximations for $\boldsymbol{D}_{R'}$ and recall that if $T \in \mathcal{G}_{\boldsymbol{D}_{R'}}$, then any $h = \mathcal{W}(T, \mathrm{Uniform}(T))$ satisfies $\mathcal{L}_{\boldsymbol{D}_{R'}}(h) \leq 1/2 - \gamma/2$. Therefore, the existence of $\mathbf{T}_{R',j^*} \in \mathcal{G}_{\boldsymbol{D}_{R'}}$, for some $j^* \in [t/R]$, implies that $\mathbf{h}_{R',j^*} \in \mathcal{H}_{R'}$ has margin at least $\gamma/2$ relative to $\boldsymbol{D}_{R'}$. Hence, Algorithm 1 can select $\mathbf{h}_{R',j^*}$ at Line 11, setting $\boldsymbol{\alpha}_{R'} = \alpha$ so that, by Claim 1, we have that $\mathbf{Z}_{R'} \leq \exp(-\gamma^2/2)$.

Now notice that, by the law of total probability,

$$\Pr\left[\mathcal{E}_{R'} \mid \cap_{r=1}^{R'-1} \mathcal{E}_r\right]$$
$$= \Pr\left[\mathrm{KL}(\boldsymbol{D}_{R'} \parallel \boldsymbol{D}_1) \leq 4\gamma^2 R \mid \cap_{r=1}^{R'-1} \mathcal{E}_r\right] \cdot \Pr\left[\mathcal{E}_{R'} \mid \cap_{r=1}^{R'-1} \mathcal{E}_r, \ \mathrm{KL}(\boldsymbol{D}_{R'} \parallel \boldsymbol{D}_1) \leq 4\gamma^2 R\right]$$
$$+ \Pr\left[\mathrm{KL}(\boldsymbol{D}_{R'} \parallel \boldsymbol{D}_1) > 4\gamma^2 R \mid \cap_{r=1}^{R'-1} \mathcal{E}_r\right] \cdot \Pr\left[\mathcal{E}_{R'} \mid \cap_{r=1}^{R'-1} \mathcal{E}_r, \ \mathrm{KL}(\boldsymbol{D}_{R'} \parallel \boldsymbol{D}_1) > 4\gamma^2 R\right]. \tag{9}$$

We will show that, conditioned on $\cap_{r=1}^{R'-1}\mathcal{E}_r$, if $\mathrm{KL}(\boldsymbol{D}_{R'}\,\|\,\boldsymbol{D}_1) \leq 4\gamma^2 R$, we can leverage Lemma 2.5 to argue that with probability at least $1 - \delta/R$ there exists a $\gamma/2$-approximation for $\boldsymbol{D}_{R'}$ within $\mathbf{T}_{R',1},\ldots,\mathbf{T}_{R',t/R}$, and that $\mathcal{E}_{R'}$ follows. On the other hand, if $\mathrm{KL}(\boldsymbol{D}_{R'}\,\|\,\boldsymbol{D}_1) > 4\gamma^2 R$, we shall prove that $\mathcal{E}_{R'}$ necessarily holds. Under those two claims, Eq. (9) yields that

$$
\begin{aligned}
\Pr\Big[\mathcal{E}_{R'} \,\Big|\, \cap_{r=1}^{R'-1}\mathcal{E}_r\Big] &\geq \Pr\Big[\mathrm{KL}(\boldsymbol{D}_{R'}\,\|\,\boldsymbol{D}_1) \leq 4\gamma^2 R \,\Big|\, \cap_{r=1}^{R'-1}\mathcal{E}_r\Big] \cdot \left(1 - \frac{d}{R}\right) \\
&\quad + \Pr\Big[\mathrm{KL}(\boldsymbol{D}_{R'}\,\|\,\boldsymbol{D}_1) > 4\gamma^2 R \,\Big|\, \cap_{r=1}^{R'-1}\mathcal{E}_r\Big] \cdot 1 \\
&\geq \Pr\Big[\mathrm{KL}(\boldsymbol{D}_{R'}\,\|\,\boldsymbol{D}_1) \leq 4\gamma^2 R \,\Big|\, \cap_{r=1}^{R'-1}\mathcal{E}_r\Big] \cdot \left(1 - \frac{d}{R}\right) \\
&\quad + \Pr\Big[\mathrm{KL}(\boldsymbol{D}_{R'}\,\|\,\boldsymbol{D}_1) > 4\gamma^2 R \,\Big|\, \cap_{r=1}^{R'-1}\mathcal{E}_r\Big] \cdot \left(1 - \frac{d}{R}\right) \\
&= 1 - \frac{\delta}{R},
\end{aligned}
$$

which, as argued, concludes the proof.

To proceed, we ought to consider the relationships between the random variables involved. To do so, for $r \in [R]$ let $\boldsymbol{\mathcal{T}}_r = \{\mathbf{T}_{r,1},\ldots,\mathbf{T}_{r,t/R}\}$. Notice that $\boldsymbol{D}_{R'}^n$ is itself random and determined by $\boldsymbol{D}_1$, and $\boldsymbol{\mathcal{T}}_1,\ldots,\boldsymbol{\mathcal{T}}_{R'-1}$.

For the first part, let $D_1$ and $\mathcal{T}_1,\ldots,\mathcal{T}_{R'-1}$ be realizations of $\boldsymbol{D}_1$ and $\boldsymbol{\mathcal{T}}_1,\ldots,\boldsymbol{\mathcal{T}}_{R'-1}$ such that $\cap_{r=1}^{R'-1}\mathcal{E}_r$ holds and $\mathrm{KL}(D_{R'}\,\|\,D_1) \leq 4\gamma^2 R$. Notice that if there exists a $\gamma/2$-approximation for $D_{R'}$ within $\boldsymbol{\mathcal{T}}_R$, then we can choose some $\mathbf{h}_{R'} \in \mathcal{H}_{R'}$ with advantage at least $\gamma/2$ so that

$$
\begin{aligned}
\prod_{r=1}^{R'} \mathbf{Z}_r = \mathbf{Z}_{R'} \cdot \prod_{r=1}^{R'-1} Z_r \\
< \mathbf{Z}_{R'} \cdot \exp(-\gamma^2(R'-1)/2) \qquad &\text{(as we condition on } \cap_{r=1}^{R'-1}\mathcal{E}_r) \\
\leq \exp(-\gamma^2 R'/2) \qquad &\text{(by Claim 1)}
\end{aligned}
$$

and, thus, $\mathcal{E}_{R'}$ follows. That is,

$$
\Pr\Big[\mathcal{E}_{R'} \,\Big|\, \cap_{r=1}^{R'-1}\mathcal{E}_r,\, \mathrm{KL}(D_{R'}\,\|\,D_1) \leq 4\gamma^2 R\Big] \geq \Pr_{\mathbf{T}_{R',1},\ldots,\mathbf{T}_{R',t/R} \overset{\mathrm{iid}}{\sim} D_1^n}\Big[\exists j \in [t/R],\, \mathbf{T}_{R',j} \in \mathcal{G}_{D_{R'}}\Big].
$$
$$
\tag{10}
$$

Finally, since by Remark 1 the distributions $D_{R'}$ and $D_1$ must have the same support, and we assume that $\mathrm{KL}(D_{R'}\,\|\,D_1) \leq 4\gamma^2 R$, Lemma 2.5 ensures that

$$
\Pr_{\mathbf{T}\sim D_1^n}[\mathbf{T} \in \mathcal{G}_{D_{R'}}] \geq \exp(-16C_{\mathrm{n}}dR).
$$

Therefore,

$$
\begin{aligned}
\Pr_{\mathbf{T}_{R',1},\ldots,\mathbf{T}_{R',t/R} \overset{\mathrm{iid}}{\sim} D_1^n}\Big[\forall j \in [t/R],\, \mathbf{T}_{R',j} \notin \mathcal{G}_{D_{R'}}\Big] &= \left(\Pr_{\mathbf{T}\sim D_1^n}\Big[\mathbf{T} \notin \mathcal{G}_{D_{R'}}\Big]\right)^{t/R} \qquad &\text{(by IIDness)} \\
&\leq (1 - \exp(-16C_{\mathrm{n}}dR))^{t/R} \\
&\leq \exp\left(-\frac{t}{R} \cdot \exp(-16C_{\mathrm{n}}dR)\right) \\
&\leq \frac{\delta}{R},
\end{aligned}
$$

where the second inequality follows since $1 + x \leq e^x$ for all $x \in \mathbb{R}$ and the last from the hypothesis that $t \geq R \cdot \exp(16C_{\mathrm{n}}dR) \cdot \ln(R/\delta)$. Considering the complementary event and applying Eq. (10), we obtain that $\mathcal{E}_{R'}$ holds with probability at least $1 - \delta/R$.

For the second part, consider instead $D_1$ and $\mathcal{T}_1,\ldots,\mathcal{T}_{R'-1}$ realizations of $\boldsymbol{D}_1$ and $\boldsymbol{\mathcal{T}}_1,\ldots,\boldsymbol{\mathcal{T}}_{R'-1}$ such that $\cap_{r=1}^{R'-1}\mathcal{E}_r$ holds and

$$
4\gamma^2 R < \mathrm{KL}(D_{R'}\,\|\,D_1),
\tag{11}
$$

and argue that $\mathcal{E}_{R'}$ necessarily follows.

Observe that

$$
\begin{aligned}
\mathrm{KL}(D_{R'} \parallel D_1) &= \mathrm{KL}(D_{R'} \parallel D_1) - \mathrm{KL}(D_{R'} \parallel D_{R'}) \\
&= \mathrm{KL}(D_{R'} \parallel D_1) - \mathrm{KL}(D_{R'} \parallel D_2) \\
&\quad + \mathrm{KL}(D_{R'} \parallel D_2) - \mathrm{KL}(D_{R'} \parallel D_3) \\
&\quad + \cdots \\
&\quad + \mathrm{KL}(D_{R'} \parallel D_{R'-1}) - \mathrm{KL}(D_{R'} \parallel D_{R'}) \\
&= \sum_{r=1}^{R'-1} \mathrm{KL}(D_{R'} \parallel D_r) - \mathrm{KL}(D_{R'} \parallel D_{r+1}).
\end{aligned}
\tag{12}
$$

Moreover, given $r \in \{1, \ldots, R'-1\}$,

$$
\begin{aligned}
\mathrm{KL}(D_{R'} \parallel D_r) - \mathrm{KL}(D_{R'} \parallel D_{r+1}) &= \sum_{i=1}^{m} D_{R'}(i) \ln \frac{D_{R'}(i)}{D_r(i)} - \sum_{i=1}^{m} D_{R'}(i) \ln \frac{D_{R'}(i)}{D_{r+1}(i)} \\
&= \sum_{i=1}^{m} D_{R'}(i) \ln \frac{D_{r+1}(i)}{D_r(i)} \\
&= \sum_{i=1}^{m} D_{R'}(i) \ln \frac{\exp(-\alpha_r c(x_i) h_r(x_i))}{Z_r} \\
&= -\ln Z_r - \sum_{i=1}^{m} D_{R'}(i) \alpha_r c(x_i) h_r(x_i).
\end{aligned}
$$

Applying it to Eqs. (11) and (12) yields that

$$
4\gamma^2 R < \mathrm{KL}(D_{R'} \parallel D_1) = -\ln \prod_{r=1}^{R'-1} Z_r - \sum_{r=1}^{R'-1} \alpha_r \sum_{i=1}^{m} D_{R'}(i) c(x_i) h_r(x_i).
$$

Thus, either

$$
-\ln \prod_{r=1}^{R'-1} Z_r > \frac{4\gamma^2 R}{2},
\tag{13}
$$

or

$$
-\sum_{r=1}^{R'-1} \alpha_r \sum_{i=1}^{m} D_{R'}(i) c(x_i) h_r(x_i) > \frac{4\gamma^2 R}{2}.
\tag{14}
$$

We proceed to analyze each case.

If Eq. (13) holds, then

$$
\prod_{r=1}^{R'-1} Z_r < \exp(-2\gamma^2 R)
$$

$$
\leq \exp(-\gamma^2 R'/2)
$$

and $\mathcal{E}_{R'}$ follows by noting that $\mathbf{Z}_{R'} = 1$ regardless of the outcome of Line 11 so $\prod_{r=1}^{R'} Z_r \leq \prod_{r=1}^{R'-1} Z_r$.

On the other hand, if Eq. (14) holds, then, letting $\mathcal{R} = \{r \in [R'-1] \mid \alpha_r \neq 0\}$,

$$
\begin{aligned}
2\gamma^2 R &< -\sum_{r=1}^{R'-1} \alpha_r \sum_{i=1}^{m} D_{R'}(i) c(x_i) h_r(x_i) \\
&= -\sum_{r \in \mathcal{R}} \alpha \sum_{i=1}^{m} D_{R'}(i) c(x_i) h_r(x_i).
\end{aligned}
$$

Since $|\mathcal{R}| \leq R$, we obtain that

$$\sum_{r \in \mathcal{R}} \frac{1}{|\mathcal{R}|} \sum_{i=1}^{m} D_{R'}(i) c(x_i)(-h_r(x_i)) > \frac{2\gamma^2}{\alpha}$$

so that there exists $h^* \in \{-h_r \mid r \in \mathcal{R}\}$ such that

$$\sum_{i=1}^{m} D_{R'}(i) c(x_i) h^*(x_i) > \frac{2\gamma^2}{\alpha}. \tag{15}$$

Moreover, from the definition of $\alpha$,

$$\begin{aligned}
\alpha &= \frac{1}{2} \ln \frac{1/2 + \gamma/2}{1/2 - \gamma/2} \\
&= \frac{1}{2} \ln \left( 1 + \frac{2\gamma}{1-\gamma} \right) \\
&\leq \frac{\gamma}{1-\gamma} \\
&< 2\gamma, \tag{16}
\end{aligned}$$

where the last inequality holds for any $\gamma \in (0, 1/2)$. Applying it to Eq. (15) yields that

$$\begin{aligned}
\sum_{i=1}^{m} D_{R'}(i) c(x_i) h^*(x_i) &> \frac{2\gamma^2}{2\gamma} \\
&\geq \gamma,
\end{aligned}$$

thus $\mathcal{L}_{D_{R'}}(h^*) < 1/2 - \gamma/2$ and, as before, $\mathcal{E}_{R'}$ follows by Claim 1 and the conditioning on $\cap_{r=1}^{R'-1} \mathcal{E}_r$. $\square$

### B.3 Proof of Theorem 2.1

**Theorem 2.1.** *There exists universal constant $C_n \geq 1$ such that for all $0 < \gamma < 1/2$, $R \in \mathbb{N}$, concept $c \colon \mathcal{X} \to \{-1, 1\}$, and hypothesis set $\mathcal{H} \subseteq \{-1, 1\}^{\mathcal{X}}$ of VC-dimension $d$, Algorithm 1 given an input training set $S \in \mathcal{X}^m$, a $\gamma$-weak learner $\mathcal{W}$,*

$$p \geq \frac{4 \ln m}{\gamma^2 R}, \qquad and \qquad t \geq e^{16 C_n d R} \cdot R \ln \frac{pR}{\delta},$$

*produces a linear classifier $\mathbf{g}$ at Line 21 such that with probability at least $1 - \delta$ over the randomness of Algorithm 1, $\mathbf{g}(x) c(x) \geq \gamma/8$ for all $x \in S$.*

*Proof.* Let $k \in \{0, 1, \ldots, p-1\}$. Applying Lemma 2.6 with failure probability $\delta/p$, we obtain that with probability at least $1 - \delta/p$,

$$\prod_{r=1}^{R} \mathbf{Z}_{kR+r} < \exp(-\gamma^2 R/2).$$

Thus, by the union bound, the probability that this holds for all $k \in \{0, 1, \ldots, p-1\}$ is at least $1 - \delta$.

Under this event, we have that

$$\begin{aligned}
\sum_{i=1}^{m} \exp\left( -c(x_i) \sum_{j=1}^{pR} \boldsymbol{\alpha}_j \mathbf{h}_j(x_i) \right) &= m \prod_{j=1}^{pR} \mathbf{Z}_j && \text{(by Claim 2)} \\
&= m \prod_{k=0}^{p-1} \prod_{r=1}^{R} \mathbf{Z}_{kR+r} \\
&\leq m \prod_{k=0}^{p-1} \exp(-\gamma^2 R/2) \\
&= m \exp(-\gamma^2 pR/2). \tag{17}
\end{aligned}$$

Now, let $\theta \geq 0$. If $c(x)\mathbf{g}(x) < \theta$, then, by the definition of $\mathbf{g}$ at Line 21, it must hold that $c(x)\sum_{j=1}^{pR}\boldsymbol{\alpha}_j\mathbf{h}_j(x) < \sum_{j=1}^{pR}\boldsymbol{\alpha}_j\theta$, thus the difference $\sum_{j=1}^{pR}\boldsymbol{\alpha}_j\theta - c(x)\sum_{j=1}^{pR}\boldsymbol{\alpha}_j\mathbf{h}_j(x)$ is strictly positive. Taking the exponential, we obtain that, for all $x \in S$,

$$\mathbf{1}_{\{c(x)\mathbf{g}(x)<\theta\}} \leq 1$$

$$< \exp\left(\sum_{j=1}^{pR}\boldsymbol{\alpha}_j\theta - c(x)\sum_{j=1}^{pR}\boldsymbol{\alpha}_j\mathbf{h}_j(x)\right)$$

$$\leq \exp(pR\alpha\theta)\exp\left(-c(x)\sum_{j=1}^{pR}\boldsymbol{\alpha}_j\mathbf{h}_j(x)\right). \qquad \text{(as } \boldsymbol{\alpha}_j \leq \alpha)$$

Therefore,

$$\sum_{i=1}^{m}\mathbf{1}_{\{c(x_i)\mathbf{g}(x_i)<\theta\}} < \exp(pR\alpha\theta)\sum_{i=1}^{m}\exp\left(-c(x_i)\sum_{j=1}^{pR}\boldsymbol{\alpha}_j\mathbf{h}_j(x_i)\right).$$

Applying Eq. (17), we obtain that

$$\sum_{i=1}^{m}\mathbf{1}_{\{c(x_i)\mathbf{g}(x_i)<\theta\}} < m\exp(pR\alpha\theta)\exp(-\gamma^2 pR/2)$$

$$= m\exp\left(pR(\alpha\theta - \gamma^2/2)\right).$$

Finally, since $0 \leq \alpha \leq 2\gamma$ (see Eq. (16)), we have that, for $0 \leq \theta \leq \gamma/8$,

$$\alpha\theta - \gamma^2/2 \leq 2\gamma \cdot \gamma/8 - \gamma^2/2$$

$$\leq -\gamma^2/4$$

and thus

$$\sum_{i=1}^{m}\mathbf{1}_{\{c(x_i)\mathbf{g}(x_i)<\gamma/8\}} < m\exp\left(-pR\gamma^2/4\right)$$

$$\leq m \cdot m^{-1} \qquad \text{(as } p \geq 4R^{-1}\gamma^{-2}\ln m)$$

$$= 1,$$

and we can conclude that all points have a margin greater than $\gamma/8$. $\qquad \square$

## C   Lower Bound

In this section, we prove Theorem 1.2. Theorem 1.2 is a consequence of the following Theorem C.2. Before we state Theorem C.2 we will: state the assumptions that we make in the lower bound for a learning algorithm $\mathcal{A}$ with parallel complexity $(p, t)$, the definition of a $\gamma$-weak learner in this section and describe the hard instance. For this let $c\colon \mathcal{X} \to \{-1, 1\}$ denote a labelling function. Furthermore, throughout Appendix C let $C_{size} := C_s \geq 1$, $C_{bias} := C_b \geq 1$ and $C_{loss} := C_l \geq 1$ denote the same universal constants.

**Assumption C.1.** Let $\mathbf{Q}^i$ with $|\mathbf{Q}^i| \leq t$ be the queries made by a learning algorithm $\mathcal{A}$ with parallel complexity $(p, t)$ during the $i$th round. We assume that a query $\mathbf{Q}_j^i \in \mathbf{Q}^i$ for $i = 1, \ldots, p$ and $j = 1, \ldots, t$ is on the form $(\mathbf{S}_j^i, c(\mathbf{S}_j^i), \mathbf{D}_j^i)$, where the elements in $\mathbf{S}_j^i$ are contained in $\mathbf{S}$, and that the distribution $\mathbf{D}_j^i$ has support $\mathrm{supp}(\mathbf{D}_j^i) \subset \{(\mathbf{S}_j^i)_1, \ldots, (\mathbf{S}_j^i)_m\}$. Furthermore, we assume $\mathbf{Q}^1$ only depends on the given sample $\mathbf{S} \in \mathcal{X}^m$ and the sample labels $c(\mathbf{S})$ where $c(\mathbf{S})_i = c(\mathbf{S}_i)$, and that $\mathbf{Q}^i$ for $i = 2, \ldots, p$ only depends on the label sample $\mathbf{S}, c(\mathbf{S})$ and the previous $i - 1$ queries and the responses to these queries.

We now clarify what we mean by a weak learner in this section.

**Definition 2.** A $\gamma$-weak learner $\mathcal{W}$ acting on a hypothesis set $\mathcal{H}$, takes as input $(S, c(S), D)$, where $S \in \mathcal{X}^* = \cup_{i=1}^{\infty}\mathcal{X}^i$, $c(S)_i = c(S_i)$ and $\mathrm{supp}(D) \subseteq \{S_1, S_2 \ldots\}$. The output of $h = \mathcal{W}(\mathcal{H})(S, c(S), D)$ is such that $\sum_i D(i)\mathbf{1}\{h(i) \neq c(i)\} \leq 1/2 - \gamma$.

We now define the hard instance which is the same construction as used in Lyu et al. [2024] (which was inspired by Karbasi and Larsen [2024]). For $d \in \mathbb{N}$, samples size $m$, and $0 < \gamma < \frac{1}{4C_b}$ we consider the following hard instance

1. The universe $\mathcal{X}$ we take to be $[2m]$.

2. The distribution $\mathcal{D}$ we will use on $[2m]$ will be the uniform distribution $\mathcal{U}$ over $[2m]$.

3. The random concept $\mathbf{c}$ that we are going to use is the uniform random concept $\{-1, 1\}^{2m}$, i.e. all the labels of $\mathbf{c}$ are i.i.d. and $\Pr_{\mathbf{c}}[\mathbf{c}(i) = 1] = 1/2$ for $i = 1, \ldots, m$.

4. The random hypothesis set will depend on the number of parallel rounds $p$, a scalar $R \in \mathbb{N}$, and the random concept $\mathbf{c}$, thus we will denote it $\mathcal{H}_{p,\mathbf{c},R}$. We will see $\mathcal{H}_{p,\mathbf{c},R}$ as a matrix where the rows are the hypothesis so vectors of length $2m$, where the $i$th entry specifies the prediction the hypothesis makes on element $i \in [2m]$. To define $\mathcal{H}_{p,\mathbf{c},R}$ we first define two random matrices $\mathcal{H}_u$ and $\mathcal{H}_{\mathbf{c}}$. $\mathcal{H}_u$ is a random matrix consisting of $R \lceil \exp(C_s d) \rceil$ rows, where the rows in $\mathcal{H}_u$ are i.i.d. with distribution $\mathbf{r} \sim \{-1, 1\}^{2m}$ ($\mathbf{r}$ has i.i.d. entries $\Pr_{\mathbf{r} \sim \{-1,1\}^{2m}}[\mathbf{r}(1) = 1] = 1/2$). $\mathcal{H}_c$ is a random matrix with $R$ rows, where the rows in $\mathcal{H}_c$ are i.i.d. with distribution $\mathbf{b} \sim \{-1, 1\}^{2m}_{C_b}$, meaning the entries of $\mathbf{b}$ are independent and has distribution $\Pr_{\mathbf{b} \sim \{-1,1\}^{2m}_{C_b}}[\mathbf{b}(i) \neq c(i)] = 1/2 - C_b\gamma$ (so $C_b\gamma$ biased towards the sign of $c$). We now let $\mathcal{H}^1_u, \mathcal{H}^1_{\mathbf{c}}, \ldots, \mathcal{H}^p_u, \mathcal{H}^p_{\mathbf{c}}$ denote i.i.d. copies of respectively $\mathcal{H}_u$ and $\mathcal{H}_{\mathbf{c}}$, and set $\mathcal{H}_{p,\mathbf{c},R}$ to be these i.i.d. copies stack on top of each other and $\mathcal{H}_{p,\mathbf{c},R} \cup \mathbf{c}$ to be the random matrix which first rows are $\mathcal{H}_{p,\mathbf{c},R}$ and its last row is $\mathbf{c}$,

$$\mathcal{H}_{p,\mathbf{c},R} = \begin{bmatrix} \mathcal{H}^1_u \\ \mathcal{H}^1_{\mathbf{c}} \\ \vdots \\ \mathcal{H}^p_u \\ \mathcal{H}^p_{\mathbf{c}} \end{bmatrix} \qquad \mathcal{H}_{p,\mathbf{c},R} \cup \mathbf{c} = \begin{bmatrix} \mathcal{H}_{p,\mathbf{c},R} \\ \mathbf{c} \end{bmatrix}.$$

5. The algorithm $\mathcal{W}$ which given matrix/hypothesis set $M \in \mathbb{R}^\ell \times \mathbb{R}^{2m}$ (where $M_{i,\cdot}$ denotes the $i$th row of $M$) is the following algorithm $\mathcal{W}(M)$.

---

**Algorithm 2:** $\mathcal{W}(M)$

**Input** : Triple $(S, c(S), D)$ where $S \in [2m]^*$, $c(S)_i = c(S_i)$ and probability distribution $D$ with $\text{supp}(D) \subset \{S_1, S_2, \ldots, \}$.
**Output:** Hypothesis $h = M_{i,\cdot}$ for some $i = 1, \ldots, \ell$ such that:
$\quad \sum_i D(i)\mathbf{1}\{\mathbf{h}(i) \neq c(i)\} \leq 1/2 - \gamma$.

1 **for** $i \in [\ell]$ **do**
2    **if** $\sum_j D(j)\mathbf{1}\{M_{i,j} \neq c(j)\} \leq 1/2 - \gamma$ // Notice that $\mathcal{W}$ doesn't know $c$ but
     can calculate this quantity using the information in $(S, c(S), D)$
     which is given as input.
3    **then**
4      **return** $M_{i,\cdot}$.
5 **return** $M_{1,\cdot}$.

---

We notice that with this construction, we have that $|\mathcal{H}_{p,\mathbf{c},R}| \leq R\lceil \exp(C_s d) \rceil + Rp$ and $\mathcal{W}(\mathcal{H}_{p,\mathbf{c},R} \cup \mathbf{c})$ a weak learner since it either finds a row in $\mathcal{H}_{p,\mathbf{c},R}$ with error less than $1/2 - \gamma$ for a query or outputs $\mathbf{c}$ which has 0 error for any query - this follows by the Assumption C.1 that the learning algorithm given $(S, \mathbf{c}(S))$ make queries which is consistent with $\mathbf{c}$.

With these definitions and notation in place, we now state Theorem C.2, which Theorem 1.2 is a consequence of.

**Theorem C.2.** *For $d \in \mathbb{N}$, $m \in \mathbb{N}$, margin $0 < \gamma < \frac{1}{4C_b}$, $R, p, t \in \mathbb{N}$, universe $[2m]$, $\mathcal{U}$ the uniform distribution on $[2m]$, and $\mathbf{c}$ the uniform concept on $[2m]$ any learning algorithm $\mathcal{A}$ with parallel complexity $(p, t)$, given labelled training set $(\mathbf{S}, \mathbf{c}(\mathbf{S}))$, where $\mathbf{S} \sim \mathcal{U}^m$, and query access to $\mathcal{W}(\mathcal{H}_{p,c,R} \cup \mathbf{c})$ we have that*

$$\mathbb{E}_{\mathbf{S},\mathbf{c},\mathcal{H}}[\mathcal{L}^{\mathbf{c}}_{\mathcal{U}}(\mathcal{A}(\mathbf{S}, \mathbf{c}(\mathbf{S}), \mathcal{W}(\mathcal{H}_{p,\mathbf{c},R} \cup \mathbf{c})))]$$
$$\geq \frac{\exp(-C_l C_b^2 \gamma^2 Rp)}{4C_l} \left( 1 - \exp\left( -\frac{m \exp(-C_l C_b^2 \gamma^2 Rp)}{8C_l} \right) - pt \exp(-Rd) \right),$$

We now restate and give the proof of Theorem 1.2.

**Theorem 1.2.** *There is a universal constant $C \geq 1$ for which the following holds. For any $0 < \gamma < 1/C$, any $d \geq C$, any sample size $m \geq C$, and any weak-to-strong learner $\mathcal{A}$ with parallel complexity $(p, t)$, there exists an input domain $\mathcal{X}$, a distribution $\mathcal{D}$, a concept $c\colon \mathcal{X} \to \{-1, 1\}$, and a $\gamma$-weak learner $\mathcal{W}$ for $c$ using a hypothesis set $\mathcal{H}$ of VC-dimension $d$ such that if the expected loss of $\mathcal{A}$ over the sample is no more than $m^{-0.01}$, then either $p \geq \min\{\exp(\Omega(d)), \Omega(\gamma^{-2} \ln m)\}$, or $t \geq \exp(\exp(\Omega(d)))$, or $p \ln t = \Omega(\gamma^{-2} d \ln m)$.*

*Proof of Theorem 1.2.* Fix $d \geq 1$, sample size $m \geq (e80C_l)^{100}$, margin $0 < \gamma \leq \frac{1}{4C_b}$, $p$ such that $p \leq \min\left\{\exp(d/8), \frac{\ln\left(m^{0.01}/80C_l\right)}{2C_l C_b^2 \gamma^2}\right\}$, $t \leq \exp(\exp(d)/8)$ and $p \ln(t) \leq \frac{d \ln\left(m^{0.01}/80C_l\right)}{8C_l C_b^2 \gamma^2}$. We now want to invoke Theorem C.2 with different values of $R$ depending on the value of $p$. We consider 2 cases. Firstly, the case

$$\frac{\ln\left(m^{0.01}/80C_l\right)}{2C_l C_b^2 \gamma^2 \lfloor \exp(d) \rfloor} \leq p \leq \frac{\ln\left(m^{0.01}/80C_l\right)}{2C_l C_b^2 \gamma^2}.$$

In this case one can choose $R \in \mathbb{N}$ such that $1 < R \leq \lfloor \exp(d) \rfloor$ and

$$\frac{\ln\left(m^{0.01}/80C_l\right)}{2C_l C_b^2 \gamma^2 R} \leq p \leq \frac{\ln\left(m^{0.01}/80C_l\right)}{2C_l C_b^2 \gamma^2 (R-1)}.$$

Let now $R$ be such. We now invoke Theorem C.2 with the above parameters and get

$$\mathbb{E}_{\mathbf{S}, \mathbf{c}, \mathcal{H}}[\mathcal{L}_{\mathcal{U}}^{\mathbf{c}}(\mathcal{A}(\mathbf{S}, \mathbf{c}(\mathbf{S}), \mathcal{W}(\mathcal{H}_{p, \mathbf{c}, R} \cup \mathbf{c})))] \tag{18}$$
$$\geq \frac{\exp(-C_l C_b^2 \gamma^2 Rp)}{4C_l}\left(1 - \exp\left(-\frac{m \exp(-C_l C_b^2 \gamma^2 Rp)}{8C_l}\right) - pt \exp\left(-Rd\right)\right),$$

We now bound the individual terms on the right-hand side of Eq. (18). Firstly, since

$$p \leq \frac{\ln\left(m^{0.01}/80C_l\right)}{2C_l C_b^2 \gamma^2 (R-1)} \leq \frac{\ln\left(m^{0.01}/80C_l\right)}{C_l C_b^2 \gamma^2 R},$$

we get that $\frac{\exp(-C_l C_b^2 \gamma^2 Rp)}{4C_l} \geq 20m^{-0.01}$ which further implies that

$$\exp\left(-\frac{m \exp(-C_l C_b^2 \gamma^2 Rp)}{8C_l}\right) \leq \exp(-10m^{0.99}) \leq e^{-10}.$$

We further notice that for $R$ as above we have that $p \ln(\exp(Rd/4)) \geq \frac{d \ln\left(m^{0.01}/80C_l\right)}{8C_l C_b^2 \gamma^2}$. This implies that $t \leq \exp(Rd/4)$, since else we would have $t > \exp(Rd/4)$ and $p \ln(t) > p \exp(Rd/4) \geq \frac{d \ln\left(m^{0.01}/80C_l\right)}{8C_l C_b^2 \gamma^2}$ which is a contradiction with our assumption that $p \ln(t) \leq \frac{d \ln\left(m^{0.01}/80C_l\right)}{8C_l C_b^2 \gamma^2}$. Since we also assumed that $p \leq \exp(d/8)$ we have that $pt \leq \exp(d/8 + Rd/4 \cdot)$. Combining this with $R > 1$ and $d \geq 1$ we have that $pt \exp\left(Rd\right) \leq \exp\left(Rd/2\right) \geq e^{-1}$. Combining the above observations we get that the right-hand side of Eq. (18) is at least

$$\mathbb{E}_{\mathbf{S}, \mathbf{c}, \mathcal{H}}[\mathcal{L}_{\mathcal{U}}^{\mathbf{c}}(\mathcal{A}(\mathbf{S}, \mathbf{c}(\mathbf{S}), \mathcal{W}(\mathcal{H}_{p, \mathbf{c}, R} \cup \mathbf{c})))] \geq 20m^{-0.01}\left(1 - e^{-10} - e^{-1}\right) \geq m^{-0.01}.$$

Now in the case that

$$p < \frac{\ln\left(m^{0.01}/80C_l\right)}{2C_l C_b^2 \gamma^2 \lfloor \exp(d) \rfloor}, \tag{19}$$

we choose $R = \lfloor \exp(d) \rfloor$. Invoking Theorem C.2 again give use the expression in Eq. (18) (with the parameter $R = \lfloor \exp(d) \rfloor$ now) and we again proceed to lower bound the right-hand side of Eq. (18). First we observe that by the upper bound on $p$ in Eq. (19), $R = \lfloor \exp(d) \rfloor$ and $\exp(-x/2) \geq \exp\left(-x\right)$ for $x \geq 1$ we get that $\frac{\exp(-C_l C_b^2 \gamma^2 Rp)}{4C_l} \geq \frac{\exp(-\ln\left(m^{0.01}/80C_l\right)/2)}{4C_l} \geq 20m^{-0.01}$, which further implies that $\exp\left(-\frac{m \exp(-C_l C_b^2 \gamma^2 Rp)}{8C_l}\right) \leq e^{-10}$. Now since $\lfloor x \rfloor \geq x/2$ for $x \geq 1$, $R = \lfloor \exp(d) \rfloor$ and we assumed that $t \leq \exp(\exp(d)/8)$ and $p \leq \exp(d/8)$ we get that

$pt \exp(-Rd) \leq \exp\left(\exp(d)/8 + d/8 - d\exp(d)/2\right) \leq \exp\left(-d\exp(d)/4\right) \leq e^{-e/4}$. Combining the above observations we get that the right-hand side of Eq. (18) is at least

$$\mathbb{E}_{\mathbf{S},\mathbf{c},\mathcal{H}}[\mathcal{L}_{\mathcal{U}}^{\mathbf{c}}(\mathcal{A}(\mathbf{S},\mathbf{c}(\mathbf{S}),\mathcal{W}(\mathcal{H}_{p,\mathbf{c},R} \cup \mathbf{c})))] \geq 20m^{-0.01}\left(1 - e^{-10} - e^{-e/4}\right) \geq m^{-0.01}.$$

Thus, for any of the above parameters $d, m, \gamma, p, t$ in the specified parameter ranges, we have that the expected loss of $\mathcal{A}$ over $\mathbf{S}, \mathbf{c}, \mathcal{H}_{p,\mathbf{c},R}$ is at least $m^{-0.01}$, so there exists concept $c$ and hypothesis $\mathcal{H}$ such that the expected loss of $\mathcal{A}$ over $\mathbf{S}$ is at least $m^{-0.01}$. Furthermore, if $\mathcal{A}$ were a random algorithm Yao's minimax principle would give the same lower bound for the expected loss over $\mathcal{A}$ and $\mathbf{S}$ as the above bound holds for any deterministic $\mathcal{A}$.

Now as remarked on before the proof the size of the hypothesis set $\mathcal{H}_{p,\mathbf{c},R}$ is at most $|\mathcal{H}_{p,\mathbf{c},R}| \leq R\lceil\exp\left(C_s d\right)\rceil + Rp$, see Item 4. Combining this with us in the above arguments having $p \leq \exp(d/8)$, $R \leq \exp(d)$ we conclude that $|\mathcal{H} \cup c| \leq \exp(\tilde{C}d/2)$ for $\tilde{C}$ large enough. Thus, we get at bound of $\log_2(|\mathcal{H} \cup c|) \leq \log_2(\exp(\tilde{C}d/2)) \leq \tilde{C}d$ which is also an upper bound of the VC-dimension of $\mathcal{H} \cup c$. Now redoing the above arguments with $d$ scaled by $1/\tilde{C}$ we get that the VC-dimension of $\mathcal{H} \cup c$ is upper bounded by $d$ and the same expected loss of $m^{-0.01}$. The constraints given in the start of the proof with this rescaling of $d$ is now $d \geq \tilde{C}$, $m \geq (e80C_l)^{100}$, $0 < \gamma \leq \frac{1}{4C_b}$, $p \leq \min\left\{\exp(d/(8\tilde{C})), \frac{\ln\left(m^{0.01}/(80C_l)\right)}{2C_l C_b^2 \gamma^2}\right\}$, $t \leq \exp\left(\exp\left(d/\tilde{C}\right)/8\right)$ and $p\ln(t) \leq \frac{d\ln\left(m^{0.01}/80C_l\right)}{8\tilde{C}C_l C_b^2 \gamma^2}$.

Thus, with the universal constant $C = \max\left\{(e80C_l)^{100}, 4C_b, \tilde{C}\right\}$ and $m, d \geq C$ and $\gamma \leq 1/C$ we have that the expected loss is at least $m^{-0.01}$ when $p \leq \min\left\{\exp(O(d)), O(\ln(m)/\gamma^2)\right\}$, $t \leq \exp\left(\exp\left(O(d)\right)\right)$ and $p\ln(t) \leq O(d\ln(m)/\gamma^2)$ which concludes the proof. $\qquad\square$

We now move on to prove Theorem C.2. For this, we now introduce what we will call the extension of $\mathcal{A}$ which still terminates if it receives a hypothesis with loss more than $1/2 - \gamma$. We further show two results about this extension one which says that with high probability we can replace $\mathcal{A}$ with its extension and another saying that with high probability the loss of the extension is large, which combined will give us Theorem C.2.

6. The output of the extension $\mathcal{B}_{\mathcal{A}}$ of $\mathcal{A}$ on input $(S, c(S), \mathcal{W})$ is given through the outcome of recursive query sets $Q^1, \ldots,$ where each of the sets contains $t$ queries. The recursion is given in the following way: Make $Q^1$ to $\mathcal{W}$ as $\mathcal{A}$ would have done on input $(S, c(S), \cdot)$ (this is possible by Assumption C.1 which say $Q^1$ is a function of only $(S, c(S))$). For $i = 1, \ldots, p$ such that for all $j = 1, \ldots, t$ it is the case that $\mathcal{W}(Q_j^{i-1})$ has loss less than $1/2 - \gamma$ under $D_j^{i-1}$ let $Q^i$ be the query set $Q$ that $\mathcal{A}$ would have made after having made query sets $Q^1, \ldots, Q^{i-1}$ and received hypothesis $\{\mathcal{W}(Q_j^l)\}_{(l,j)\in[i-1]\times[t]}$. If this loop ends output the hypothesis that $\mathcal{A}$ would have made with responses $\{\mathcal{W}(Q_j^l)\}_{(l,j)\in[i]\times[t]}$ to its queries. If there is an $l, j$ such that $\mathcal{W}(Q_j^l)$ return a hypothesis with loss larger than $1/2 - \gamma$ return the all 1 hypothesis.

We now go to the two results we need in the proof of Theorem C.2. The first result Corollary 1 says that there exists an event $E$ which happens with high probability over $\mathcal{H}_{p,\mathbf{c},R}$ such that $\mathcal{A}$ run with $\mathcal{W}(\mathcal{H}_{p,\mathbf{c},R}, \mathbf{c})$ is the same as $\mathcal{B}_{\mathcal{A}}$ run with $\mathcal{W}(\mathcal{H}_{p,\mathbf{c},R})$. This corollary can be proved by following the proofs of Theorem 5 and 8 in Lyu et al. [2024] and is thus not included here.

**Corollary 1.** *For $d \in \mathbb{N}$, $m \in \mathbb{N}$, margin $0 < \gamma < \frac{1}{4C_b}$, labelling function $c : [2m] \to \{-1, 1\}$, $R, p, t \in \mathbb{N}$, random matrix $\mathcal{H}_{p,c,R}$, learning algorithm $\mathcal{A}$, $\mathcal{B}_{\mathcal{A}}$, training sample $S \in [2m]^m$, we have that there exist and event $E$ over outcomes of $\mathcal{H}_{p,c,R}$ such that*

$$\mathcal{A}(S, c(S), \mathcal{W}(\mathcal{H}_{p,c,R} \cup c))\mathbf{1}_E = \mathcal{B}_{\mathcal{A}}(S, c(S), \mathcal{W}(\mathcal{H}_{p,c,R}))\mathbf{1}_E$$

*and*

$$\Pr_{\mathcal{H}_{p,c,R}}[E] \geq 1 - pt\exp\left(-Rd\right).$$

The second result that we are going to need is Lemma C.3 which relates parameters $R, \beta, p$ to the success of any function of $(S, \mathbf{c}(S), \mathcal{H}_{p,\mathbf{c},R})$ which tries to guess the signs of $\mathbf{c}$ — which is the number of failures in our hard instance. For a training sample $S \in [2m]^*$ we will use $|S|$ to denote the number of distinct elements in $S$ from $[2m]$, so for $S \in [2m]^m$ we have $|S| \leq m$.

**Lemma C.3.** *There exists universal constant $C_s, C_l \geq 1$ such that: For $m \in \mathbb{N}$, $p \in \mathbb{N}$, $\mathcal{H}_{p,\beta,\mathbf{c},R}$, function $\mathcal{B}$ that takes as input $S \in [2m]^m$ with labels $\mathbf{c}(S)$, and hypothesis set $\mathcal{H}_{p,\beta,\mathbf{c},R}$, we have that*

$$\Pr_{\mathbf{c},\mathcal{H}_{p,\mathbf{c},R}} \left[ \sum_{i=1}^{2m} \mathbf{1}\{\mathcal{B}(S,\mathbf{c}(S),\mathcal{H}_{p,\mathbf{c},R})(i) \neq \mathbf{c}(i)\} \geq \frac{(2m-|S|)\exp(-C_l C_b^2 \gamma^2 Rp)}{2C_l} \right]$$

$$\geq 1 - \exp\left( -\frac{(2m-|S|)\exp(-C_l C_b^2 \gamma^2 Rp)}{8C_l} \right).$$

We postpone the proof of Lemma C.3 and now give the proof of Theorem C.2.

*Proof of Theorem C.2.* We want to lower bound $\mathbb{E}_{\mathbf{S},\mathbf{c},\mathcal{H}}[\mathcal{L}_{\mathcal{U}}^{\mathbf{c}}(\mathcal{A}(\mathbf{S},\mathcal{W}(\mathcal{H} \cup \mathbf{c})))]$. To this end since $\mathbf{S}$ and $\mathbf{c}$ are independent and $\mathcal{H}_{p,\mathbf{c},R}$ depended on $\mathbf{c}$ the expected loss can be written as

$$\mathbb{E}_{\mathbf{S},\mathbf{c},\mathcal{H}}[\mathcal{L}_{\mathcal{U}}^{\mathbf{c}}(\mathcal{A}(\mathbf{S},\mathbf{c}(\mathbf{S}),\mathcal{W}(\mathcal{H}_{p,\mathbf{c},R} \cup \mathbf{c})))]$$
$$= \mathbb{E}_{\mathbf{S}}\left[\mathbb{E}_{\mathbf{c}}\left[\mathbb{E}_{\mathcal{H}_{p,\mathbf{c},R}}[\mathcal{L}_{\mathcal{U}}^{\mathbf{c}}(\mathcal{A}(\mathbf{S},\mathbf{c}(\mathbf{S}),\mathcal{W}(\mathcal{H}_{p,\mathbf{c},R} \cup \mathbf{c})))]\right]\right].$$

Now let $S \in [2m]^m$, $c$ be any outcome of $\mathbf{S}$ and $\mathbf{c}$. Then for these $S,c$ we have by Lemma C.3 that there exists some event $E$ over $\mathcal{H}_{p,c,R}$ such that

$$\mathcal{L}_{\mathcal{U}}^{c}(\mathcal{A}(S,c(S),\mathcal{W}(\mathcal{H}_{p,c,R},c)))\mathbf{1}_E = \mathcal{L}_{\mathcal{U}}^{c}(\mathcal{B}_{\mathcal{A}}(S,c(S),\mathcal{W}(\mathcal{H}_{p,c,R})))\mathbf{1}_E,$$

and

$$\Pr_{\mathcal{H}_{p,c,R}}[E] \geq 1 - pt\exp(-Rd),$$

furthermore, define $E'$ be the event that

$$E' = \left\{ \sum_{i=1}^{2m} \mathbf{1}\{\mathcal{B}_{\mathcal{A}}(S,c(S),\mathcal{W}(\mathcal{H}_{p,c,R}))(i) \neq c(i)\} \geq \frac{(2m-|S|)\exp(-C_l C_b^2 \gamma^2 Rp)}{2C_l} \right\}.$$

Using the above and $\mathcal{U}$ being the uniform measure on $[2m]$ so assigns $1/(2m)$ mass to every point and that $|S| \leq m$ we now get that

$$\mathbb{E}_{\mathcal{H}_{p,c,R}}[\mathcal{L}_{\mathcal{U}}^{c}(\mathcal{A}(S,c(S),\mathcal{W}(\mathcal{H}_{p,c,R},c)))] \geq \mathbb{E}_{\mathcal{H}_{p,c,R}}[\mathcal{L}_{\mathcal{U}}^{c}(\mathcal{A}(S,c(S),\mathcal{W}(\mathcal{H}_{p,c,R},c)))\mathbf{1}_E\mathbf{1}_{E'}]$$
$$= \mathbb{E}_{\mathcal{H}_{p,c,R}}[\mathcal{L}_{\mathcal{U}}^{c}(\mathcal{B}_{\mathcal{A}}(S,c(S),\mathcal{W}(\mathcal{H}_{p,c,R})))\mathbf{1}_E\mathbf{1}_{E'}]$$
$$\geq \frac{(2m-|S|)\exp(-C_l C_b^2 \gamma^2 Rp)}{4C_l m}\mathbb{E}_{\mathcal{H}_{p,c,R}}[\mathbf{1}_E\mathbf{1}_{E'}]$$
$$\geq \frac{\exp(-C_l C_b^2 \gamma^2 Rp)}{4C_l}\left(1 - \Pr_{\mathcal{H}_{p,c,R}}[\overline{E'}] - pt\cdot e^{-Rd}\right).$$

We can do this for any pair $c$ and $S \in [2m]^m$, so we have that

$$\mathbb{E}_{\mathbf{c}}\left[\mathbb{E}_{\mathcal{H}_{p,\mathbf{c},R}}[\mathcal{L}_{\mathcal{U}}^{\mathbf{c}}(\mathcal{A}(S,\mathbf{c}(S),\mathcal{W}(\mathcal{H}_{p,\mathbf{c},R} \cup \mathbf{c})))]\right]$$
$$\geq \frac{\exp(-C_l C_b^2 \gamma^2 Rp)}{4C_l}\left(1 - \Pr_{\mathbf{c},\mathcal{H}_{p,\mathbf{c},R}}[\overline{E'}] - pt\exp(-Rd)\right).$$

Now by Lemma C.3 and $|S| \leq m$ we have that $\Pr_{\mathbf{c},\mathcal{H}_{p,\mathbf{c},R}}\left[\overline{E'}\right]$ is at most

$$\Pr_{\mathbf{c},\mathcal{H}_{p,\mathbf{c},R}}\left[\overline{E'}\right] \leq \exp\left(-\frac{(2m-|S|)\exp(-C_l C_b^2 \gamma^2 Rp)}{8C_l}\right) \leq \exp\left(-\frac{m\exp(-C_l C_b^2 \gamma^2 Rp)}{8C_l}\right).$$

I.e. we have shown that

$$\mathbb{E}_{\mathbf{c}}\left[\mathbb{E}_{\mathcal{H}_{p,\mathbf{c},R}}[\mathcal{L}_{\mathcal{U}}^{\mathbf{c}}(\mathcal{A}(S,\mathbf{c}(S),\mathcal{W}(\mathcal{H}_{p,\mathbf{c},R} \cup \mathbf{c})))]\right]$$
$$\geq \frac{\exp(-C_l C_b^2 \gamma^2 Rp)}{4C_l}\left(1 - \exp\left(-\frac{m\exp(-C_l C_b^2 \gamma^2 Rp)}{8C_l}\right) - pt\exp(-Rd)\right),$$

for any $S \in [2m]^m$. Now by taking expectation over $\mathbf{S} \sim \mathcal{U}^m$ we get that

$$\mathbb{E}_{\mathbf{S},\mathbf{c},\mathcal{H}}[\mathcal{L}_{\mathcal{U}}^{\mathbf{c}}(\mathcal{A}(\mathbf{S},\mathbf{c}(\mathbf{S}),\mathcal{W}(\mathcal{H}_{p,\mathbf{c},R} \cup \mathbf{c})))]$$
$$\geq \frac{\exp(-C_l C_b^2 \gamma^2 Rp)}{4C_l}\left(1 - \exp\left(-\frac{m\exp(-C_l C_b^2 \gamma^2 Rp)}{8C_l}\right) - pt\exp(-Rd)\right),$$

which concludes the proof. $\square$

We now prove Lemma C.3 which is a consequence of maximum-likelihood, and the following Fact 1, where Fact 1 gives a lower bound on how well one from $n$ trials of a biased $\{-1, 1\}$ random variable, where the direction of the bias itself is random, can guess this random direction of the bias.

**Fact 1.** *For function $f : \{-1, 1\}^n \to \{-1, 1\}$ and $0 < \gamma \le \frac{1}{4C_b}$*

$$\mathbb{E}_{\mathbf{c}\sim\{-1,1\}}[\mathbb{E}_{\mathbf{b}\sim\{-1,1\}^n_{C_b}}[1\{f(\mathbf{b}) \ne \mathbf{c}\}]] \ge \exp(-C_l C_b^2 \gamma^2 n)/C_l.$$

*Proof.* This is the classic coin problem. The lower bound follows by first observing, by maximum-likelihood, that the function $f^\star$ minimizing the above error is the majority function. The result then follows by tightness of the Chernoff bound up to constant factors in the exponent. $\square$

With Fact 1 in place we are now ready to proof Lemma C.3, which we restate before the proof

*Proof of Lemma C.3.* Let $\mathcal{H}_{C_b}$ be the matrix consisting of the i.i.d $C_b\gamma$ biased matrices in $\mathcal{H}_{p,c,R}$, $\mathcal{H}_{\mathbf{c}}^1, \ldots, \mathcal{H}_{\mathbf{c}}^p$ stack on top of each other,

$$\mathcal{H}_{C_b} = \begin{bmatrix} \mathcal{H}_{\mathbf{c}}^1 \\ \vdots \\ \mathcal{H}_{\mathbf{c}}^p \end{bmatrix}.$$

Furthermore, for $i = 1, \ldots, 2m$ let $\mathcal{H}_{C_b,i}$ denote the $i$th column of $\mathcal{H}_{C_b}$ which is a vector of length $pR$. Now for $i$ inside $S$, $\mathcal{B}$ has the sign of $\mathbf{c}(i)$, so the best function that $\mathcal{B}$ can be is to be equal to $\mathbf{c}(i)$. For $i$ outside $S$, $\mathcal{B}$ does not know $\mathbf{c}(i)$ from the input but has information about it through $\mathcal{H}_{C_b,i}$, we notice that the sign's of the hypotheses in $\mathcal{H}_u^1, \ldots, \mathcal{H}_u^p$ and $\mathcal{H}_{C_b,j}$ $j \ne i$ and $\mathbf{c}(S)$ is independent of $\mathbf{c}(i)$ and does not hold information about $\mathbf{c}(i)$, thus the best possible answer any $\mathcal{B}$ can make is to choose the sign which is the majority of the sign's in $\mathcal{H}_{C_b,i}$ - the maximum likelihood estimator. We now assume that $\mathcal{B}$ is this above-described "best" function - as this function will be a lower bound for the probability of failures for any other $\mathcal{B}$, so it suffices to show the lower bound for this $\mathcal{B}$. Now with the above described $\mathcal{B}$, we have that

$$X := \sum_{i=1}^{2m} \mathbf{1}\{\mathcal{B}(S, \mathbf{c}(S), \mathcal{H}_{p,\mathbf{c},R})(i) \ne \mathbf{c}(i)\} = \sum_{i\notin S} \mathbf{1}\left\{\operatorname{sign}\left(\sum_{j=1}^{pR} \mathcal{H}_{C_b\, i,j}\right) \ne \mathbf{c}(i)\right\}.$$

Thus, we have that $X$ is a sum of $2m - |S|$ (where $|S|$ is the number of distinct elements in $S$) independent $\{0, 1\}$-random variables and by Fact 1 we have that the expectation of each these random variables is at least $\mathbb{E}_{\mathbf{c},\mathcal{H}_{p,c,R}}[X] \ge (2m - |S|)\exp(-C_l C_b^2 \gamma^2 Rp)/C_l$. Thus, we now get by Chernoff that

$$\Pr_{\mathbf{c},\mathcal{H}_{p,\mathbf{c},R}}\left[X \ge \frac{(2m - |S|)\exp(-C_l C_b^2 \gamma^2 Rp)}{2C_l}\right] \ge \Pr_{\mathbf{c},\mathcal{H}_{p,\mathbf{c},R}}[X \ge \mathbb{E}[X]/2]$$

$$\ge 1 - \exp(-\mathbb{E}[X]/8) \ge 1 - \exp\left(-\frac{(2m - |S|)\exp(-C_l C_b^2 \gamma^2 Rp)}{8C_l}\right),$$

as claimed. $\square$

