# OpenReview forum: "Optimal Parallelization of Boosting"
_NeurIPS.cc/2024/Conference — NeurIPS 2024 oral_

### Official Review · Reviewer_GD3o · 2024-06-16

**Soundness:** 4
**Presentation:** 4
**Contribution:** 4
**Rating:** 8
**Confidence:** 4

**Summary:**

This paper studies parallelization in weak-to-strong boosting algorithms. Such algorithms are modeled by the number of sequential rounds $p$ that they run for, and the amount of work $t$ that can be done in parallel in each round. Each unit of work in a round is typically a query to a weak learning algorithm, that outputs a hypothesis from a class of VC dimension $d$ (and these queries can be instantiated in parallel). Formally, in round $i$, the algorithm invokes (in parallel) the weak learner with distributions $D^i_1,\dots,D^i_t$, and obtains $h^i_1,\dots,h^i_t$ such that the error of $h^i_j$ wrt $D^i_j$ is at most $1/2-\gamma$. There are $p$ such rounds, at the end of which, the weak-to-strong learning algorithm outputs some weighted vote over all the $h^i_j$s obtained so far. Ideally, we want the final classifier output by the algorithm to be *strong*: at the very least, its error should be competitive with that of AdaBoost (which is $\tilde{O}(d/m\gamma^2)$ where $m$ is the number of training samples).

Under a model of weak-to-strong learning defined as above, the classic AdaBoost works with $p=O(\ln m / \gamma^2)$, and $t=1$. What are some other reasonable tradeoffs that we can hope for?

Karbasi and Larsen (2024) gave an algorithm that works with $p=1$ and $t=\exp(O(d \ln m / \gamma^2))$. This was followed up on by Lyu et al. (2024), who obtain $p=O(\ln m/\gamma^2R)$ and $t=\exp(O(dR^2))\ln(1/\gamma)$ for any $1 \le R \le 1/2\gamma$. Both Karbasi and Larsen (2024) as well as Lyu et al. (2024) also gave some lower bounds, but neither covered the entire spectrum of $p$ and $t$ in terms of tightness with respect to algorithms achieving these bounds.

This paper largely fills up these gaps. On the upper bound side, the authors present an algorithm that achieves $p=O(\ln m/\gamma^2R)$ and $t=\exp(O(dR)) \ln \frac{\ln m}{\delta \gamma^2}$ for any $R \ge 1$. Observe that the bound on $t$ improves Lyu et al. (2024)'s bound by a factor $R$ in the exponent. The authors also show lower bounds that are tight (upto log factors) in nearly in all regimes.

Both, the algorithm for the upper bound, and the lower bound instance, are inspired by the work of Lyu et al. (2024).

### **Upper bound**

There are $p$ sequential rounds. For simplicity, we describe the first round, prior to which $D_1$ is set to the uniform distribution on the training sample. We break our computation into $R$ chunks in parallel. Each chunk, invokes a weak learner $t/R$ times in parallel on a fresh sample drawn from $D_{1}$, to obtain $t/R$ many hypotheses in total (Thus, the total number of invokations to the weak learner across all the $R$ chunks is $R \cdot t/R = t$ as required.)

Thereafter, there are $R$ sequential rounds of boosting. As $r$ ranges from $1,\dots,R$, we try to obtain a classifier that has error at most $1/2-\gamma$ with respect to $D_r$. We simply do this by checking if there was a hypothesis in the $r$th chunk that has such an error with respect to $D_r$ using the sample we had (which, notably, was from $D_1$). If we do find such a hypothesis, we do a standard boosting update to derive $D_{r+1}$. Assuming that the hypotheses in each step had the required errors with respect to $D_r$, we can imagine that each step works correctly as a standard boosting step, and hence, in each of the $p$ rounds, we are in fact doing $R$ rounds of boosting (and hence, $p$ can be a factor $R$ smaller than standard AdaBoost).


But do the hypotheses in each step have the required properties? When we have a sample from $D_1$, we can simply see if a hypothesis has error at most $1/2-\gamma$ with respect to $D_1$ by checking the error of the hypothesis on the sample itself --- this follows from standard uniform convergence of VC classes. However, what if we have a sample from $D_1$, but want to check if a hypothesis has error at most $1/2-\gamma$ with respect to $D_2$? Can we still use the empirical error on the sample as a proxy? In fact, this is what the algorithm is doing in each boosting step. Intuitively, if the distributions $D_2$ and $D_1$ are "close", this should still work. But note that we make exponential updates to $D_1$ in the boosting step, so it is not obvious at all that $D_2$ should be close to $D_1$. Lyu et al. (2024) control the max-divergence between $D_2$ and $D_1$, and show that this recipe works by using sophisticated tools like advanced composition from differential privacy. This is where the authors diverge (no pun intended): instead of the max-divergence, the authors control the KL divergence between $D_2$ and $D_1$ instead. This is acheieved by using the Gibbs variational principle. The technical analysis seems highly non-trivial, but gets the job done: with good chance over the sample, the empirical error on a sample from $D_1$ is going to be a good proxy for the distributional error on $D_2$, provided the KL divergence between $D_2$ and $D_1$ is small. If the KL is not small, then the authors show that progress has already been made. In this way, by tracking KL divergence instead of the max-divergence, the authors are able to improve over the bound of Lyu et al. (2024).

### **Lower bound**

The analysis for deriving the improved lower bound is much more involved. We first start by describing the high level construction in Lyu et al. (2024). The ground truth hypothesis is a random concept $c$ on a domain twice the size of the training set. The hypothesis class $\mathcal{H}$ that the weak learner operates over is also constructed randomly. In particular, it contains $c$, and also $p$ other hypothesis $h_1,\dots,h_p$, where each $h_i$ on each $x$ agrees with $c$ with probability $1/2+2\gamma$. The VC dimension of this class can be controlled in terms of $p$. Now, whenever the weak learner gets queried with a distribution $D$, if it can satisfy this query by returning a hypothesis that is not $c$, it does so. The goal is to argue that the weak learner can get away with never having to return $c$ at all in any round. If this is the case, what the learning algorithm knows about the rest of the domain is only in the form of $2\gamma$-biased coins. By instantiating the lower bound on learning the bias of a coin, we get a lower bound on the number of rounds.

Lyu et al. (2024) require the number of queries $t$ in each round to be sufficiently small for the weak learner to never return $c$. The main observation by the authors is that, indeed, it is possible to use a much bigger bias than $2\gamma$ in the construction of $h_i,\dots,h_p$. That is, each hypothesis can be biased towards $c$ to a much larger extent (as much as $\sqrt{\ln(m)/p} \gg 2\gamma$). This lets them relax the number of allowed queries $t$ per round, which ultimately yields the stronger lower bound.

**Strengths:**

This paper essentially completes the characterization of the tradeoff between the number of sequential rounds and the parallel work in each round in boosting algorithms. Previous work left gaps between the upper and lower bounds across much of the spectrum of these parameters. The authors improve upon the state of the art, using highly non-trivial analyis tools, and essentially close the gaps across nearly all of the spectrum. We now have a significantly more complete picture about the tradeoffs involved in parallelizing boosting thanks to the authors' work.

**Weaknesses:**

The paper "Boosting, Voting Classifiers and Randomized Sample Compression Schemes" (https://arxiv.org/pdf/2402.02976) by da Cunha et al. (2024) is a relevant paper to the present work---in particular, we can get rid of at least one of the two log factors in the error of AdaBoost with a voting classifier. I recommend the authors at least mention this and cite the paper.

I would also encourage the authors to discuss (somewhere in the paper, maybe as a separate paragraph, or in the conlusion) a bit more about the only regime that we still don't know a matching upper bound for: that of $t \ge \exp(\exp(d))$.

Minor/typos:\
Line 64: $n$ hasn't been introduced yet (it should be the size of the training set? and maybe also use $m$ then?)\
Line 132: Shellah -> Shelah

**Questions:**

Is there some intuitive meaning to the lower bound of $t \ge \exp(\exp(d))$, even at a very high level? It seems like such a bound on $t$ (albeit weaker) also existed in Lyu et al. (2024). Do you have any thoughts on how one may attempt to close it, or the inherent difficulty?

**Limitations:**

The authors adequately address any limitations that I can foresee.

---

> ### Author Rebuttal · Authors · 2024-08-05
>
> We thank the reviewer for the thoughtful evaluation of our work. It was a joy for us to read it. It is clear that the reviewer built a solid understanding of our work, grasping multiple of the subtleties in the argument. This even extends to related works to some extents, as evidenced by the reviewer's comments and the insightful suggestion of a recent reference, which we will adopt. In fact, in our opinion, such level of comprehension seems deserving of a higher confidence score.
>
> Regarding the question on why the $\exp(\exp(d))$ term appears, indeed it is somewhat unclear whether it should truly be there. Simply examining the calculations, it originates from the following argument in the lower bound: For each parallel round, we have around $N=\exp(d)$ many random hypotheses that could be used to answer the query distributions (since the VC-dimension is $d$). Since the query distributions in round $i$ are independent of the random hypotheses used to answer queries in round $i$, if each of them is a valid response with just constant probability (say $1/e$), then the chance that none of them are a valid response to a fixed query distribution is only $e^{-N} = \exp(-\exp(d))$ (recall the hypotheses are chosen randomly). So for a parallel algorithm to ask a query that forces the weak learner to return the true concept, we would need to ask around $t = \exp(\exp(d))$ queries. We acknowledge that this is not super intuitive, but at least this is where it originates. It would be very interesting to exploit this in a new algorithm.

---

> ### Comment · Reviewer_GD3o · 2024-08-07
> **Response to rebuttal**
>
> Thank you for the intuition on the $\exp(\exp(d))$ lower bound on $t$. I maintain my score of 8, and indeed, I am confident that this is a strong contribution and should be accepted (updated confidence 3 -> 4). Great work again!

---

### Official Review · Reviewer_umpM · 2024-06-28

**Soundness:** 4
**Presentation:** 3
**Contribution:** 4
**Rating:** 7
**Confidence:** 3

**Summary:**

The authors study parallelized boosting, a natural weak-to-strong learning model recently re-introduced by Larsen and Karbasi. Building on recent work of Lyu et al., this work gives new upper and lower bounds on the trade-off between number of rounds, and number of parallel calls per round to the weak learner, and in particular resolves the complexity of parallel boosting up to log factors in a certain natural parameter regime.

A boosting algorithm is a method for amplifying a `weak’ learner assumed to have some advantage $\gamma$ (that is classification accuracy $1/2+\gamma$) to a strong learner (achieving accuracy $1-\varepsilon$ with probability $1-\delta$) by repeated rounds of calls to the weak learner on sequentially modified ground distributions, typically taking a weighted majority vote of the results.

A $(p,t)$-parallelized boosting algorithm makes p rounds of t-calls to the weak learner, where each round can only depend on the outputs of previous rounds. The authors main result is a new upper giving a tradeoff between $p$ and $t$ for learning any hypothesis class $H$ with VC dimension $d$. In particular, for any $R \in \mathbb{N}$, they give a boosting algorithm with

$$p=\frac{\log(m)}{\gamma^2 R}, t=e^{dR}\log\frac{\log m}{\delta R}$$

Here $m$ is the number of samples used by the algorithm, which is assumed to achieve near-optimal accuracy-sample trade-off $m \approx \tilde{O}(d\varepsilon^{-1}\gamma^{-2})$. This improves over prior work which gave a similar result for $t=e^{dR^2}$, reducing the R-dependence from quadratic to linear in the exponent.

Second, the authors improve prior lower bounds on parallelized boosting to show their bound is near-tight in many regimes of interest. In particular, they prove that either $p \geq \min(exp(d), \log(m)\gamma^{-2}))$ or $t \geq exp(exp(d))$, or $p\log t \geq d\gamma^{-2}\log(m)$. The last of these matches the upper bound up to log factors, so the authors resolve the problem in the regime where $t < exp(exp(d))$, $p < \min(exp(d), \log(m)\gamma^{-2})$, and under the requirement of near-optimal accuracy-sample tradeoff.

**Strengths:**

Boosting is one of the most successful and broadly used paradigms in machine learning. Understanding the extent to which boosting can be parallelized is a core problem, and of great interest to the learning theory and machine learning communities. This work makes substantial progress on resolving the complexity of parallelized boosting.

The main technique introduced to improve Lyu et al.’s upper bound is a novel and elegant "win-win" theorem, analogs of which may be useful in other problems. The rough idea is to "simulate" sequential boosting distributions $D_0,\ldots,D_R$ in each round, and look at $KL(D_0,D_R)$. If the KL between these distributions is close, the authors argue that one can essentially simulate sampling from $D_R$ by sampling from $D_0$ (up to some small error), meaning the `simulated’ boosting will be successful and adopt the guarantees of standard sequential boosting. If the KL is large, we cannot simulate samples but this indicates the boosting algorithm has made progress and we win anyway. This method removes the sub-optimal $R$ factor from Lyu et al.’s method using the simpler max-divergence instead of KL-divergence.

**Weaknesses:**

My main complaint is that I feel the results are a little bit over-stated in the abstract and early in the introduction, which claims to essentially resolve the complexity of parallelized boosting. This doesn’t really seem true, since as discussed above the problem only seems to be resolved (up to log factors) under three assumptions:

1.   $t < exp(exp(d))$

2.   $p < \min \{exp(d), \log(m)\gamma^{-2})\}$

3.   The algorithm is required to have near-optimal accuracy-sample tradeoff.

Note that the latter $p$-dependence is not so restrictive, since this is achieved by non-parallel boosting (I.e. t=1), but the other parameter regimes remain open. It is unclear to me how restrictive the last condition is. It seems very reasonable one would be willing to sacrifice somewhat on samples to achieve higher parallelization; is this possible?

**Questions:**

It Is it possible one might achieve better trade-offs by relaxing the sample-optimality assumption? Or is this largely an assumption made to simplify the formulae for $p$ and $t$ in terms of samples and not accuracy.

---

> ### Author Rebuttal · Authors · 2024-08-05
>
> We thank the reviewer for the thoughtful review. As for Reviewer GD3o, we see that the present reviewer got a solid understanding of our work and its contributions. The question posed by the reviewer attests to this and, once again, we share our opinion that such level of comprehension is suitable for a higher confidence score.
>
> Answering the question, it is indeed possible to achieve better $p$ vs. $t$ tradeoffs by further relaxing the restriction on the sample complexity of the algorithm. In more detail, the $\log n$ factor in the upper and lower bounds may be replaced by $\log(1/\varepsilon)$ factors for a target accuracy of $\varepsilon$ greater than or equal to the accuracy we obtain. We chose to focus on the near-optimal accuracy regime to keep the formulas as simple as possible with the already numerous parameters.
>
> We agree with the author that emphasizing it can make the scope of our contribution clearer. We will add a discussion of this to the paper.

---

> > ### Comment · Reviewer_umpM · 2024-08-07
> > **Rebuttal Response**
> >
> > I acknowledge the authors' rebuttal. My confidence score is based on the fact that I did not check the works' math line by line, though I ensured the main claims were plausible and trust the authors' correctly proved them. I am confident in my overall assessment of the paper, and that it should be accepted.

---

### Official Review · Reviewer_D6oP · 2024-07-12

**Soundness:** 3
**Presentation:** 3
**Contribution:** 2
**Rating:** 7
**Confidence:** 3

**Summary:**

The authors offer the  bound of algorithm 1 in  paper in a very traditional learning theory.

**Strengths:**

I think a theory understanding of the algorithm is more important than the experiment reports. This paper shows the bounds for a kind of parallel boosting algorithm. The proof sturcture of algorithms is clear. Authors present their work clearly.

**Weaknesses:**

The most important problem for this work is the view of boosting and the applicability of algorithm 1.
1. After the work of XGBoost, the proof of boosting is to minimize the loss value of model on training dataset instead of the combining the weak learners. From this aspect, can we gain a better bound or design a better parallel boosting algorithm?
2. I really like the proof work in this paper, but the  fatal problem in this paper is that the algorithm 1 may be not accelerate the model training. For line 10 to line 18, the algorithm have to find $h^*$ in $H_{kR+r}$ and this process may be a exhausting work, which means algorithm may cost more time than traditional boosting with the same computing resource.

**Questions:**

Please show the importance of algorithm 1 or show that algorithm 1 can accelerate model training under the same computing resource.

**Limitations:**

The same with weakness

---

> ### Author Rebuttal · Authors · 2024-08-05
>
> We thank the reviewer for the effort invested in evaluating our submission. We were happy to see that the reviewer found the presentation of our arguments clear and values the theoretical nature of our work, which is its entire focus.
>
> We remark that with a lot of parallel computation, the time to find $h^\star$ in $H_{kR+r}$ may be reduced (that is, the time to completion, not the total work). In particular, one thread can evaluate the performance of each $h \in H_{kR+r}$ in parallel and then the best performing $h^\star$ can be computed.

---

### Decision · Program_Chairs · 2024-09-25

**Decision:**

Accept (oral)

**Comment:**

The reviewers unanimously agreed that this is a theoretically solid paper that contributes several strong upper and lower bounds for understanding parallel boosting algorithms. The meta-reviewer would be happy to recommend the paper for acceptance.